KEYLINK: towards a more integrative soil representation for inclusion in ecosystem scale models—II: model description, implementation and testing

http://orcid.org/0000-0003-2760-3015 Flores Omar 1 2 omar.flores@uantwerpen.be
http://orcid.org/0000-0003-3794-7735 Deckmyn Gaby 2
http://orcid.org/0000-0002-3221-6960 Curiel Yuste Jorge 3 4
http://orcid.org/0000-0002-6168-5467 Javaux Mathieu 5 6
Uvarov Alexei 7
van der Linde Sietse 8
http://orcid.org/0000-0001-9523-3453 De Vos Bruno 9
http://orcid.org/0000-0002-8051-8517 Vereecken Harry 6
Jiménez Juan 10
http://orcid.org/0000-0002-7060-2459 Vinduskova Olga 2
Schnepf Andrea 6
1 Biogeography and Global Change, National Museum of Natural Sciences, Consejo Superior de Investigaciones Científicas , Madrid , Spain
2 PLECO, Department of Biology, Universiteit Antwerpen , Antwerp , Belgium
3 BC3—Basque Centre for Climate Change , Leioa , Spain
4 IKERBASQUE—Basque Foundation for Science , Bilbao , Spain
5 Earth and Life Institute, Université Catholique de Louvain , Louvain-la-Neuve , Belgium
6 Agrosphere Institute (IBG-3), Forschungszentrum Jülich GmbH , Jülich , Germany
7 Laboratory of Soil Zoology, A.N. Severtsov Institute of Ecology and Evolution, Russian Academy of Sciences , Moscow , Russia
8 Forest Research , Farnham , UK
9 Department of Environment and Climate, Research Institute for Nature and Forest , Brussels , Belgium
10 Instituto Pirenaico de Ecología, Consejo Superior de Investigaciones Científicas , Jaca , Spain
Björk Robert
Electronic publication date: 2021 Jan 15
Publication date: 2021
Volume: 9
Electronic Location ID: e10707
Received 2020 Apr 7; Accepted 2020 Dec 14
Copyright: © 2021 Flores et al.
Copyright year: 2021
Copyright holder: Flores et al.
License: This is an open access article distributed under the terms of the Creative Commons Attribution License, which permits unrestricted use, distribution, reproduction and adaptation in any medium and for any purpose provided that it is properly attributed. For attribution, the original author(s), title, publication source (PeerJ) and either DOI or URL of the article must be cited.
License URL: https://creativecommons.org/licenses/by/4.0/

Keywords: Ecosystem models, Soil food web, Soil matrix, Ecosystem engineering, Hydrology, Soil organic matter, Soil structure, Predator exclusion, Trophic cascades, Growth rates

Funding: COST (European Cooperation in Science and Technology) FP1305 (BioLink) and ES1406 (KEYSOM) Short Term Scientific Mission (STSM) programs Spanish Ministry of Science, Innovation and Universities Spanish Ministry of Economy and Competitiveness (MINECO) IBERYCA (CGL2017-84723-P) BC3 María de Maeztu Excellence Accreditation MDM-2017-0714 Basque Government BERC 2018-2021 This article is based upon work from COST Actions FP1305 (BioLink) and ES1406 (KEYSOM), supported by COST (European Cooperation in Science and Technology), and their Short Term Scientific Mission (STSM) programs. Omar Flores’ work was funded by FPU PhD grant program of the Spanish Ministry of Science, Innovation and Universities. Jorge Curiel Yuste received funding from the Spanish Ministry of Economy and Competitiveness (MINECO) under projects IBERYCA (CGL2017-84723-P) and the BC3 María de Maeztu excellence accreditation (MDM-2017-0714). Jorge Curiel Yuste also received funding from the Basque Government through the BERC 2018-2021 program. The funders had no role in study design, data collection and analysis, decision to publish, or preparation of the manuscript.

==============================
New knowledge on soil structure highlights its importance for hydrology and soil organic matter (SOM) stabilization, which however remains neglected in many wide used models. We present here a new model, KEYLINK, in which soil structure is integrated with the existing concepts on SOM pools, and elements from food web models, that is, those from direct trophic interactions among soil organisms. KEYLINK is, therefore, an attempt to integrate soil functional diversity and food webs in predictions of soil carbon (C) and soil water balances. We present a selection of equations that can be used for most models as well as basic parameter intervals, for example, key pools, functional groups’ biomasses and growth rates. Parameter distributions can be determined with Bayesian calibration, and here an example is presented for food web growth rate parameters for a pine forest in Belgium. We show how these added equations can improve the functioning of the model in describing known phenomena. For this, five test cases are given as simulation examples: changing the input litter quality (recalcitrance and carbon to nitrogen ratio), excluding predators, increasing pH and changing initial soil porosity. These results overall show how KEYLINK is able to simulate the known effects of these parameters and can simulate the linked effects of biopore formation, hydrology and aggregation on soil functioning. Furthermore, the results show an important trophic cascade effect of predation on the complete C cycle with repercussions on the soil structure as ecosystem engineers are predated, and on SOM turnover when predation on fungivore and bacterivore populations are reduced. In summary, KEYLINK shows how soil functional diversity and trophic organization and their role in C and water cycling in soils should be considered in order to improve our predictions on C sequestration and C emissions from soils.

Introduction

Soil models used in ecosystem-scale modelling need to be relatively simple and fast at performing calculations. Nonetheless, carbon (C) and nutrient turnover and hydrology are extremely important for determining ecosystem productivity and C sequestration in the ecosystem. The most widely used soil models (Century, RothC) emphasize the C flow from easily degradable to stable organic compounds using first-order kinetics to describe their decay rates (Campbell & Paustian, 2015). The relevance of chemical recalcitrance, used in those models, is accepted in the early stages of litter decomposition, but that approach has been questioned on the long term soil organic matter (SOM) stabilization (Schmidt et al., 2011), highlighting the relevance of other processes in the physical protection of SOM within soil matrix (Deckmyn et al., 2020). This has led to the development of models including an explicit representation of structural effects on SOM (Kuka, Franko & Rühlmann, 2007). Furthermore, recent studies have shown that microbial products from the transformation of plant litter are the largest contributors to stable SOM (Mambelli et al., 2011; Cotrufo et al., 2013).

The insights concerning the role of the microbial biomass in C turnover has been introduced in models such as MIcrobial-MIneral Carbon Stabilization (MIMICS) model (Wieder et al., 2014, 2015) and Litter Decomposition and Leaching (LIDEL) model (Campbell et al., 2016). However, soil fauna and especially ecosystem engineers, that is, organisms that create, modify or maintain habitats by changing the physical structure of the ecosystem (Jones, Lawton & Shachack, 1994), have also been shown to play a key role in determining C and nutrient turnover and hydrology of soils through their impact on aggregation, pore formation and bioturbation as well as their direct contribution to litter and SOM turnover (Filser et al., 2016; Lavelle et al., 2016). Several authors have highlighted the need to include soil fauna contributions to SOM dynamics into soil modelling (see review by Vereecken et al. (2016)). This information has been used in detailed and small-scale soil models (Chertov et al., 2017a; Geisen et al., 2019), but is not incorporated into larger-scale ecosystem models. The main difficulty is the lack of data concerning the soil, either physical, chemical or biological, and the different methods used, making parameterization of any model unsure. The goal of the KEYLINK model is to consider the soil including the main mechanisms concerning the effects of soil biota on litter and SOM transformations and hydrology through structural modifications, without increasing the number of parameters beyond what is currently available on most well-measured ecosystems (Deckmyn et al., 2020). We show how this model has been parameterized for a forest stand where soil fauna was never studied in detail, but many other soil and stand characteristics are well established.

The core model concept is the strong link between soil biota, soil structure and turnover (Fig. 1). The decay of fresh litter is dependent on the recalcitrance and carbon to nitrogen (N) ratio (C:N) of the litter, although different soil biota groups have specific sensibilities to recalcitrance and C:N ratio. For SOM, the turnover depends on the accessibility, linked to the pore size distribution, the aeration and H2O in the pores and the aggregation (based on the model by Kuka, Franko & Rühlmann (2007)). Both SOM and litter turnover depend on temperature and humidity. Soil fauna, specifically ecosystem engineers, directly affect pore distribution besides an important effect on bioturbation. Pore distribution affects hydrology which again affects all soil processes.

Figure 1 Simplified model scheme.

General structure of the KEYLINK model. Square boxes represent pools of organic matter. Wide double-line arrows, with a circle within the arrow, represent fluxes between pools (blue arrowheads show bidirectional fluxes). Isolated circles represent abiotic factors that affect model fluxes, and red narrow arrows connect each factor or pool with the model parts (at the arrowheads) that are regulated by them.

The scientific background for the model is fully described in Deckmyn et al. (2020). Here, the related processes are formulated mathematically. Finally, we show how the model can simulate several known mechanisms of soil faunal effects such as changes in litter recalcitrance affecting fungal/bacterial ratio, changes in pH affecting earthworm populations, effects of ecosystem engineers on bioturbation and hydrology, and importance of microbivores and predators in the soil fauna food web.

Methodology

The KEYLINK model has been conceptually designed integrating the structure of the soil by its porosity, the hydrology and the C cycle through the soil food web. Those key parts of the soil interact (Fig. 1) determining the rates of SOM stabilization and CO2 emissions from soil. The functions developed to represent and simulate those processes are presented here.

Structural effects

Pore size distribution determines accessibility for trophic interactions of soil fauna and soil microorganisms (Fig. 2), both by size and by aeration and H2O; soil fauna changes pore size distribution and produces cracks and fissures in the soil. In the model, pore size distribution is divided into the following five categories:

Figure 2 Pools and fluxes in the KEYLINK model.

Scheme of C pools with their interactions. All pools, soil, microorganisms and fauna (see Table S1.2) are represented in the model in the same units (g C m−3). The arrows represent carbon fluxes between the pools; each arrow is represented by a term in the model equations.

Inaccessible pores (<0.1 µm in diameter): pores around inaccessible C (within the micro-aggregate, organo-clay interaction). Water is held here but is not available to plants (measured from wilting point). The volume of inaccessible pores is related to the clay content and type.

Bacterial pores (0.1–2 µm): the pores within macro-aggregates and pores in loam, accessible only to bacteria. Engineer saprotrophs (e.g., earthworms) can also use SOM in these pores (and in the following pore categories, all except inaccessible pores) because they eat directly all soil.

Micropores (2–30 µm): pores not accessible to macrofauna, mesofauna and most predators, but accessible to microfauna bacterivores and fungivores, fungi, mycorrhiza and bacteria. Water is held at field capacity but available to plants. In sandy soil and within macro-aggregates (>250 µm), pores fall in this category.

Mesopores (30 µm–1.5 mm): pores where most soil fauna can penetrate (not macrofauna) between large macro-aggregates (>1 mm) or formed by fine roots. Mesopore volume can be determined in the field from drained water capacity (but this includes macropores). These pores are well aerated also at field capacity, but can dry out below field capacity.

Macropores (>1.5 mm): cracks or biopores formed by ecosystem engineers. They are of vital importance for soil hydrology as preferential flow through these pores has a major impact on infiltration rate. These are the first pores to have O2 when water level is above field capacity, but dry out quickly below field capacity.

The initial values of soil porosity in the model simulations can be calculated from measured soil water retention curves, or even using models such as Saxton et al. (1986) that yield field capacity, porosity and wilting point from the C, clay and sand contents, or using measured bulk density (Db). Following Malamoud et al. (2009), the percentage of total porosity (P%) can be computed from Db and soil particle density (DS) as shown in Eq. (2). DS can be measured or is calculated from Dm = soil mineral particle density (2.65 g cm−3) and DSOM = organic particle density (1.35 g cm−3) as: (1) Ds=100%SOMDSOM+100−%SOMDm

(2) P%=Ds−DbDs100

Water flow

We advise using KEYLINK model in combination with a detailed water model including preferential flow through macropores as well as good representation for matrix flow (s.a. Richards’ equation). However, we show in this paper how it can be used with a simpler representation of water flow but still allowing the important dynamic interactions between pore sizes and hydrology that are fundamental to the model. A spilling bucket approach is used at a daily time-step, where water drains from a layer into the underlying layer when its water content is above field capacity in the soil matrix. However, in contrast to conventional spilling bucket models, we allow water to flow faster through macropores (before the soil matrix is saturated). Net precipitation (Pnet) is calculated as:(3) Pnet=P−E

where P is precipitation (mm day−1) and E is evapotranspiration (mm day−1) from measured or modelled data (vegetation model). Infiltration (I) is assumed to be equal to the part of precipitation entering the soil. Infiltration and runoff (Prunoff, mm day−1) must equal Pnet.

(4) I+Prunoff=Pnet

Infiltration is composed of water entering the soil matrix, water filling the macropores and water draining from macropores. Water that enters macropores remains in the macropore domain or enters the layers below. The fraction of infiltration entering macropores depends on the surface area of the macropores (SAmacro), assumed cylindrical. Assume measured or derived maximal infiltration rate (ImaxMat, mm day−1) of the soil matrix. Maximal infiltration rate through macropores (ImaxPor, mm day−1) is calculated from the volume of the pores (PVmacro), assumed not limiting at daily scale, plus infiltration capacity of the layer (n + 1) in which the macropores end.

(5) ImaxPor=PVmacro+ImaxMat(n+1)

If Pnet > (ImaxPor + ImaxMat) runoff is calculated as: (6) Prunoff=Pnet−(ImaxPor+ImaxMat)

after which calculations continue using Pnet − Prunoff as net precipitation.

If ImaxMat < Pnet < (ImaxPor + ImaxMat) the soil matrix is filled at a rate equal to the maximum infiltration rate, all other water is lost either through the macropores to the next layer or by filling macropores. If ImaxMat > Pnet the soil matrix is filled with water, traditional spilling bucket, but an equivalent volume is lost through macropores to the bottom layer depending on the surface area of the macropores. The total soil water volume of soil layer n, SWn, is then limited by the total pore volume of the layer and the water already filling the pores, and is calculated as:(7) SWn=SWn+min(PVn−SWn,Imaxmat(1−SAmacro),Pnet(1−SAmacro))

For drainage (D) to the bottom layer, the spilling bucket approach is used plus a portion of water that goes straight through the macropores, calculated from the surface area of the macropores.

(8) Dn=PnetSAmacro+Pnet−min(PVn−SWn,Imaxmat(1−SAmacro),Pnet(1−SAmacro))

For each pore size class the fraction water filled is calculated from the water content: so always one pore size is partially saturated and all others are either saturated or dry within one layer.

C flow

The KEYLINK model combines SOM modelling with soil food web modelling. The model conceptualized in Fig. 2 has 13 carbon pools (Table S1.2), visualised by boxes. Above and belowground litter is assumed to be provided from an external source (tree shoot in Fig. 2) not covered by this model. It could be given through experimental data or an external model, for example, a tree growth model that delivers the input of litter into the litter pool. All simulations presented here were made with constant C inputs (Table S2.6). Exudation is an input of organic carbon released from roots into the SOM pool. Every live pool has a respiration rate (r) and a turnover or death rate (d). On consuming a C pool, a fraction of this pool always becomes faeces and enters the SOM pool except for the microbial pools, that is, microbes and microbivores. SOM can be distributed in different fractions, particulate organic matter (POM) and dissolved organic matter (DOM), which can gain relevance in the addition and simulation of other nutrient cycles and processes as leaching. However, here, as a first version of the model, we present a simplification using SOM as a uniform pool. The growth (G, g C m−3 day−1) of a biomass pool (B, g C m−3) is described according to Monod kinetic,

(9) G=∑n=1N⁡(gmax(SfaKs+S)n)B

where gmax (g C g C−1 day−1) is the maximal rate of growth, to which several modifiers are applied (see descriptions below). Substrate (S, g C m−3) is the consumable pool, litter, SOM or biomass of soil organism (n), that consumer pool (B) can use but corrected by its available fraction (fa). All fluxes (N) of consumed C from each S are summed. Ks (g C m−3) is related to substrate quality, it gives the content required to get half the maximal growth. This is not related to the amount that will be consumed, because consumed C equals growth + faeces, but shows how dense the material needs to be ‘found’ by the consumer. Available fraction of a S to a consumer (as fa) is calculated using the fraction from total porosity volume that is accessible for the consumer, by size, minus its fraction that is completely flooded or dry (see Eqs. (14)–16). This availability introduces the concept of physical recalcitrance, highlighting the role that soil structure plays affecting C fluxes in the soil, because SOM decomposition rates modelling use to rely on its chemical recalcitrance, from now on referred just as “recalcitrance”. But physical recalcitrance has proven to be also relevant for the calculation of SOM decomposition rates (Von Lützow et al., 2008), and soil matrix also affect other biotic interactions through the food web by this availability concept.

Rate of increase of a population of meso- or macrofauna depends on generation time (r, K strategies), age distribution of the population, different life stages. Models exist for only some soil fauna species (Osler & Sommerkorn, 2007; Chertov et al., 2017a). To offer a solution that can work for both the microbial biomass and the meso- and macrofauna, we use gmax as equal to the maximal rate of increase in biomass of any population, dB/dt = gmax when resources are non-limiting and assuming the population structure is stable and optimal, equal to what is often stated as the intrinsic growth rate of a species (Birch, 1948).

The net rate of change of a biomass pool is the sum of growth (G), respiration (R) and turnover (death, Dt), and possibly predation (Pd), all in g C m−3 day−1:(10) dBdt=G−R−Dt−Pd

R is a function of temperature, through respiration rate (r, g C g C−1 day−1), and biomass, assuming the same temperature sensitivity as growth; this is somewhat different to how it is seen in many models where a food source is turned over with a specific efficiency. From a more faunal point of view, this makes sense: a food source is “consumed”; the consumed material is partly excreted and partly assimilated and spent on respiration and growth (i.e., biomass formation).

(11) R=rB

While the death rate d (g C g C−1 day−1) is constant.

(12) Dt=dB

Predation depends on biomass of predator or microbivore and is calculated from the growth of the predator (Gpred, from Eq. (9)) plus the fraction of the prey allocated to faeces (ffaec).

(13) Pdprey=Gpred(1+ffaec)

Effect of H2O

Drought or saturation of a pore leads to reduced availability of the C in the pore for its food web consumers. First, the overall effect of hydration is calculated as a modifier (mH2Otot) in function of volumetric soil moisture (V) and pore volume (Pvol) (after Freytag & Luttich, 1985).

(14) mH2Otot={4VPvol(1−VPvol)forVPvol<0.51forVPvol>0.5

The activity is always in the pores that are not water-logged, therefore the pore size class that is partially filled with water, and the pore size above that is assumed not yet completely dry (after Kuka, Franko & Rühlmann, 2007).

(15) mH2O=PvolAPvolA+PvolWmH2Ototfortheporespartiallyfilled,

(16) mH2O=PvolWPvolA+PvolWmH2Ototfortheporesoneclassabove,

where PvolW is the water filled pore volume and PvolA is the aerated pore volume. The availability (fa) of a substrate to a consumer is defined by the inherent availability of the pore size to the consumer, multiplied with mH2O. For surface litter these calculations are not possible since the surface litter is not in the soil matrix. However, on days without precipitation, litter humidity is assumed to be related to the soil humidity below, therefore the mH2O calculated for the microbial biomass is used.

Simulating the variability in gmax

The maximum growth of biota is influenced by different environmental factors. Each one can lead to a modifier (m ∈ [0, 1]) on gmax. It is easy to change, add or turn off specific modifiers according to the soil studied. Here we present a modelling framework focused on abiotic controls of growth rates, but there is room for new add-ons as for example death rate modifiers as a density-dependent microbial turnover. While interaction processes affected by the demographic density of microbial communities (e.g., competition, space constraints) can play also a significant role controlling growth and decomposition rates and improve its modelling (Georgiou et al., 2017), our aim in this work is to link the key roles of fauna and soil structure in C cycle modelling, and together with the hydrology can simulate constraints in biotic interactions, which are also relevant controls in microbial growth and activity.

Simulating the effect of temperature (T)

To simulate the effect of T on growth rate through a temperature modifier (mT), we use a Q10-shaped curve between maximum tolerable temperature (Tmax) and minimum temperature for consumers activity (Tmin), set as a default at 0 °C (Franko, 1989), but unlike many models, we assume a plateau above the optimal temperature (Topt).

(17) mT={0,T<TminorT≥TmaxQ(T−Topt)/10,Tmin≤T<Topt1,Topt≤T<Tmax

However, temperature also increases respiration (R). To simulate this temperature effect, we assume the same Q10 function but without the plateau; in this way, when T is above the optimum, R increases while growth does not. At some point these lines will cross and cause a net reduction in biomass.

Effect of pH on growth

A good example of an optional effect is the effect of pH: for a system close to a threshold, simulating pH can be very important, assuming a good knowledge of the system. But for well-buffered systems, it is an unnecessary increase in complexity. gmax decreases at low pH for bacteria but increases for fungi (Rousk, Brookes & Bååth, 2009, 2010; Rousk et al., 2010). For this example, we put the thresholds at 8 for fungi and 3 for bacteria, with precision of one decimal, inducing a 10 fold reduction in gmax for a change in pH of 1.

(18) mpH=1/((pH–8)10)forfungiifpH≥8.1

(19) mpH=1/((3−pH)10)forbacteriaifpH≤2.9

In any other case for bacteria or fungi, mpH = 1, and if mpH goes above 1, it is replaced by 1. For engineer saprotrophs, their optimal gmax changes (becoming gmaxEng) with pH (Lavelle, Chauvel & Fragoso, 1995; Chertov et al., 2017b) according to the following equation:(20) gmaxEng={0,ifpH<3(gmax2)(pH−3),if3≤pH<5gmax,ifpH≥5

Effect of recalcitrance and C:N on gmax

Overall consumption of an organism that can consume different pools is computed by simply adding them up. However, litter is not necessarily as “palatable”, depending on its C:N ratio if not enough N, and on recalcitrance if low in energy, then it is needed to consume more litter, which is calculated through modifiers mC:N and mrec, respectively. This is simulated by changing gmax. The equation for mrec is not necessary and only important if enough data on litter quality is available or the users are interested into looking into the effects of changes in litter quality. The litter pool can be consumed by both bacteria and fungi, and of course also detritivores. Depending on the C:N ratio, the competition between these two is different; this is simulated by the gmax of the bacteria being more variable with C:N ratio. The sensitivity is described by the parameter pmC:N, between 0 and 1.

(21) fungi:mC:Nfung=min(1,(C:NfungC:Nlit)pmC:Nfung)

(22) bacteria:mC:Nbact=min(1,(C:NbactC:Nlit)pmC:Nbact)

For litter recalcitrance (Reclit), a linear equation instead of a power is chosen so that decay of the recalcitrant litter is 0 if pmRec = 1 and is unaffected if pmRec = 0. The reason for choosing a different equation than above is that constrain of labile litter decomposition by C:N ratio should not completely stop decomposition but adjust the decomposition rate, while recalcitrant fraction of litter could remain almost constant for long periods. The following equations determine if the recalcitrant fraction of litter remains stable or if it is affected by decomposers partially or even totally.

(23) fungi:mrecfung=min(1,1−pmRecfungReclit)

(24) bacteria:mrecbact=min(1,1−pmRecbactReclit)

Adding up all these modifying effects on gmax

We assume a complete additivity of the effects, so the different modifiers on gmax are multiplied to get the overall effect, mtot in Eq. (25). Another optional approach could be to use only the most limiting effect, setting mtot equal to the lowest modifier and ignoring the rest.

(25) mtot=mTmC:NmrecmpHmH2O

Not assimilated C

The reduction in a substrate equals the growth of the consumer plus the C that goes to faeces (excrements) and to respiration. Fraction to excrement (ffaec) is a parameter of the consumer and assumed constant. However, one consumes more and a larger fraction becomes faeces at a lower substrate quality, for the meso-and macro fauna, because microbes do not produce excrements; the sensitivity of ffaec to C:N ratio is expressed by the modifier mfaec. This is however only relevant for the detritivores and engineers (Eq. (26)) that eat SOM and litter which can contain extremely variable amounts of nutrients; for the predators and herbivores we assume the variability is minimal. For the microbes, it was calculated as an effect on gmax.

(26) ffaecEff=ffaec+mfaecC:NSOM−C:NengC:NSOMffaec

Calculations regarding engineers

Soil changes made by engineer species depend on their body width, but in the model this is simplified using initial parameters for engineers’ effects that must be chosen based on an average width (see Table S2.5); the model then simulates their daily effects using their biomasses. Bioturbation is a function of engineer biomass (Beng, g C m−3), which calculates organic matter moving to deeper layers: litter moving (g Clit g Ceng−1 day−1) from litter layer to end of burrow, and SOM moving by mixing of soil between layers (g CSOM g Ceng−1 day−1). In this first version of the model, with only one soil layer, bioturbation works as a C output flow, but in future versions with more layers it could be upgraded to C flows between them.

Burrow volume (PVB, l m−3) is a function of engineer biomass (g Ceng m−3) and the ratio of pore volume to engineer biomass (VEratio, l g Ceng−1) but towards a maximum (PVBmax, l m−3). On the other hand, burrow turnover (tPVB) happens at a constant rate, with average burrow lifespan of 10 years; porosity decreases and burrows become mesopores.

(27) PVB=max(min(PVBmax,VEratioBeng),(PVmacro−PVtextmacro)(1−tPVB))

where PVmacro is the current pore volume of the macropores and PVtextmacro is the textural porosity of the macropores (see in next section). This could be improved in future versions including perturbations as the possible effect of heavy rain.

Porosity calculations

The pore volume is distributed in five classes by pore size. Initial pore size distribution is given or measured as the total pore volume (PV, l m−3) in each class. The link between aggregation and porosity is hard to quantify. Regelink et al. (2015a) showed for different soils that overall soil porosity is the sum of the textural porosity determined by the proportion of clay, sand and silt fractions and aggregation porosity. They conclude that micropores, which they define <9 µm, are mainly situated within the aggregates, while mesopores are situated between dry-sieved aggregates. While Regelink et al. (2015a) have shown that total micro and mesoporosity (<1,000 µm) increases with total aggregate content, Grosbellet et al. (2011) have provided evidence that pores in the range 30–300 µm decrease with aggregation. Despite of the generally lower ranges for mesopores (9–1,000 µm) described for soil physics (Lal & Shukla, 2004; Regelink et al., 2015a), here mesopores are assumed to be physically accessible to mesofauna body size (ca. 100–2,000 µm), so we consider that mesopores ranging 30–1,500 µm are a reasonable compromise. Based on that, we decided to hypothesize that aggregation increases bacterial and micro- porosity while decreasing mesoporosity. However, we want to emphasize that further experimental studies are needed to establish robust relationships between aggregation and pore size distribution.

Aggregates are not calculated as a pool, but the effect of aggregation is included in the calculation of porosity as described below. The following three porosities contribute to total porosity:Textural porosity (PVtext): measured or calculated from % clay and sand.

Additional aggregation porosity (PVAg): all porosity in surplus of textural, can be estimated, for example from PTF (PedoTransfer Function) or calculated empirically from SOM and fungal biomass, that is, mycorrhiza and other fungi, max 2% porosity extra (Eq. (29)). Aggregation (Ag) is the fraction (0–1) of the SOM aggregated calculated as (based on the data from Malamoud et al., 2009):

(28) Ag=min(1,c(Bfung+Bmyc)BSOM)

with an empirical parameter c = 10. The aggregation porosity is then calculated as:(29) PVAg=kAgBSOM

with k = coefficient (2 l g C−1 m−3) based on empirical data (Regelink et al., 2015a, 2015b).Bioporosity (PVB): biopores created by engineers. Pore formation by engineers increases macroporosity, increasing soil layer density, but at the same time reduces mesoporosity as engineers push soil aside and produce casts that are denser than average soil. The relative importance of these two effects depends on the engineers’ activity patterns, and is reflected by the parameter fPV ∈ (0, 1), which gives the fraction of the change in biopore volume that increases macroporosity. Therefore, the counterpart of the biopore volume (1 − fPV) PVB is ‘compensated’ by a decrease in mesoporosity.

Conceptually, the total soil porosity is then the sum of:(30) PVtot=PVtext+PVAg+fPVPVB

In the model, pore volume is calculated for each pore size separately.

The volume of micropores (PVmicro) and bacterial pores (PVbact) increases with increasing aggregation. Apart from creating additional porosity depending on the total amount of aggregated SOM (Eq. (29)), aggregation also increases the relative micro- and bacterial pore volume at the expense of (textural) mesoporosity (PVmeso), therefore not increasing total porosity. This effect is controlled by available pore space between mineral particles (i.e., textural mesoporosity) and we assume that half of this mesoporosity can be affected by aggregation. In both cases, we assume that the increase in porosity due to aggregation is divided equally among micropores and bacterial pores. The pore volume in different size classes is calculated as:(31) PVmacro=PVtextmacro+PVB

(32) PVmeso=PVtextmeso−(1−fPV)PVB−Ag2PVtextmeso

(33) PVmicro=PVtextmicro+kAg2BSOM+Ag4PVtextmeso

(34) PVbact=PVtextbact+kAg2BSOM+Ag4PVtextmeso

Volume of inaccessible pores is assumed to be constant and equal to PVtextinac.

These changes are calculated daily to give a dynamic feedback to the hydrology and to the distribution of each C source among pore classes, affecting its availability.

Leaching

Water leaving one soil layer (n) is moved to the layer below (n + 1). Dissolved organic and inorganic compounds are a complex matter to simulate since they are strongly dependent on the pH and the mother-material, that is, clay and Ca rich or not. Nonetheless, in many systems simulating leaching of N and DOM is highly relevant. Unless better data are available, we suggest the following, semi-empirical method:

Dissolved organic matter can be simulated in relation to the CO2 released as total respiration (Rtot, g C m−3 day−1) based on the consideration that high “activity” in the soil is related to high Rtot. This is calculated as a fraction (fDOM) of Rtot entering the DOM pool, similar to the concepts used in the LIDEL model, in addition to the directly exuded DOM (CExud, g C m−3 day−1) which is an input (from data or a vegetation model). Assuming a short half-life of DOM and semi-empirically, because daily concentration is not “equal” to daily production (DOMp, g C m−3 day−1) but is linearly related to the daily production, we consider:(35) DOMp=CExud+fDOMRtot

DOM has a short half-life but the dissolution is even faster (hours). We assume the daily concentration is in equilibrium between dissolved and adsorbed (DOMad) depending on adsorption coefficient KD of the soil (m3 kg−1 soil). Similar to the modelling in Orchidee-SOM (Camino-Serrano et al., 2018) we assume:(36) DOMad=KDDOM

with KD depending on the minerals and pH. More clay (fClay fraction) means less mobile DOM, and lower pH is also a cause of less mobile DOM.

(37) KD=aKD−bKDpH+cKDfClay

with values 0.001226, 0.000212 and 0.00374 respectively for aKD, bKD and cKD, from Camino-Serrano et al. (2018).

DOM leaching is calculated as the volume of water leaching to a lower layer multiplied with the concentration of dissolved DOM.

Calculation order

The sequence of function sets used by the model to calculate all carbon fluxes and ecosystem changes is as follows:

Calculate the pore size fractions in 5 classes and the associated pore surface areas

Calculate the water volume of the relevant pore size

Use the precipitation leaching to calculate DOM leaching

Calculate for each biota group the accessibility of each of the pools it consumes

Calculate the gmax depending on temperature, H2O, C:N, pH and recalcitrance

Solve the 12 differential equations for increase/decrease of all C pools

Update all C pools

Calculate the new C:N and recalcitrance of each pool

Calculate engineering effecti1) Update macropores

i2) Update SOM from bioturbation

Calculate other changes in pore size distribution from weather or management

The KEYLINK core model consists of steps d to i; steps a, b, c and j are add-ons that could be replaced by other models (e.g., water flow model) coupled to KEYLINK. Steps a–c are used to calculate the distribution of porosity between the pore classes, the hydrology and daily soil water content (distributed among pore classes), and then step d calculates how that is affecting the availability of each C source to its consumers. That couples soil structure and hydrology with trophic interactions, allowing the resolution of differential equations for C fluxes.

Model coding and output

KEYLINK consists of a relatively limited, freely downloadable Python code (available at: https://github.com/Plant-Root-Soil-Interactions-Modelling/KEYLINK). Each of the modifiers on growth, that is, temperature, pH, H2O, recalcitrance and C:N, as well as the primal shape of the growth equations can be adapted towards specific questions or ecosystems. The inputs in the current version are read from data-files but are easy to link to a mechanistic model. The output of the current version consists of all daily C pools as well as the main C fluxes. KEYLINK is also available as a stand-alone executable model, allowing it to be called from models in other languages. A single run of ten years could take less than one minute (depending on computing power). In this version, the average results over one hundred runs are calculated but also all daily outputs of each run are saved.

Model parameterization

The first version of KEYLINK model has been parameterized for a Scots pine forest stand situated in Brasschaat, in the Campine region in Belgium (51°18′ N and 4°31′ E) but without modelling the complete forest ecosystem (to simplify the interpretation of the results from KEYLINK). The goal of this parameterization was a model verification, not a model application for which a complete integration with an aboveground model or detailed above ground data would be necessary.

The soil of the Brasschaat forest is sandy but with high ground water table, so trees are generally not water-limited, but the topsoil is often dry. The soil is acidic (pH 3.5). The trees were planted around 1930 and formed a rather sparse vegetation in 1999, with leaf area index (LAI) ranging from 2.1 to 2.4.

For this model run, we used the following input data from the stand (Table 1). In this case, we did not use measured or modelled growing trees but constant input of aboveground and belowground litter (measured value). The top 90 cm of soil from the Brasschaat forest was analyzed in Janssens et al. (1999). Earthworm biomass, used in this case as an example of ecosystem engineers, is extremely low due to the low pH, it was not measured since 1993 by Muys (1993), but these data are used since there is no reason to expect there was a marked change.

Table 1 Initial input data.

Data from Brasschaat Scots pine forest (Belgium). Microbial C pool was estimated as hot water extractable C (HWC).

Variable	Unit	Value	Reference	
Earthworm biomass	g C m−3	200	Muys (1993)	
pH		3.5	Janssens et al. (1999)	
Sand	%	93	Janssens et al. (1999)	
Initial SOM	g C m−3	11,470	Janssens et al. (1999)	
Initial litter	g C m−3	2,680	Janssens et al. (1999)	
Fine root biomass	g C m−3	400	Janssens et al. (2002)	
Fine root litter	g C m−3	300	Janssens et al. (1999)	
Fine root growth rate	g C m−3 year−1	210	Janssens et al. (2002)	
Annual litter fall	g C m−3 year−1	400	Horemans et al. (2017)	
Fine root turnover	g C m−3 year−1	740	Based on Janssens et al. (2002)	
C input to mycorrhiza	g C m−3 year−1	197	Assumed based on Deckmyn et al. (2014)	
Microbial C as HWC	g m−3	1,338.21	Gaublomme, De Vos & Cools (2006)	

Data availability on soil pools, biology and functioning is generally low, and it is currently not possible to find a dataset describing in detail, and with small error margins, the temporal evolution of all different soil biological compartments and SOM pools. Available data are often incomplete, or based on rough estimates, for example, from semiquantitative DNA analysis for microbial abundance in soils. To deal with this issue, a quite pragmatic approach combining different estimates from different sources is appropriate for most datasets where the soil is not the key focus, but a means to improve the simulation of an ecosystem.

The daily loss of water by evapotranspiration was calculated using an equation for potential evapotranspiration based on Thornthwait (1948) in this study.

Model calibration

Once the model is parameterized for an ecosystem, the next step is to optimize that model, calibrating the fit of its simulations to the ecosystem data. The optimization included in the KEYLINK model follows a Bayesian procedure, using the version of Markov Chain Monte Carlo (MCMC) method known as the Metropolis-Hastings random walk with reflection algorithm (Christian & Casella, 1999; Van Oijen, Rougier & Smith, 2005; Van Oijen, 2008).

A pragmatic assumption is that the starting values of the C pools (including the soil fauna initial biomasses) are at steady state for a given date (most often spring or summer, it would be unrealistic to keep the values constant through the year as they fluctuate with climate). The simplest calibration of any ecosystem can be done by assuming these 11 carbon pools (litter, SOM and the nine functional groups in food web) need to be stable over the simulated years, for example, for 9 years that gives us 99 data points by taking the same value for each C pool every year (Table 2). Initial litter, SOM and biomasses of bacteria, fungi and engineers were taken from the references cited in Table 1. For other C pools, data were estimated using measured data for previous C pools and similar proportions between C pools as in the Swedish pine forest in Persson et al. (1980); predator biomass was assumed to be the 20% of all biomass in their consumed C pools. Errors were assumed as a percentage of biomass, 10% for predators, 12.5% for litter and SOM and 20% for the rest C pools.

Table 2 Calibration data.

Data of C pools used for the model calibration. Biomasses of the nine food web functional groups: bacteria (Bb), fungi (Bf), mycorrhiza (Bmyc), bacterivores (Bbvores), fungivores (Bfvores), detritivores (Bdet), engineers (Beng), herbivores (Bhvores) and predators (Bpred); and the other two C pools: litter and soil organic matter (SOM). Values were used once per year during calibration at days 180, 545, 910, 1,275, 1,640, 2,005, 2,370, 2,735 and 3,100.

C pool	Value (g C m−3)	Error (g C m−3)	
Bb	15.1	3.02	
Bf	15.1	3.02	
Bmyc	160	32	
Bbvores	0.1	0.02	
Bfvores	0.8	0.16	
Bdet	0.6	0.12	
Beng	0.2	0.04	
Bhvores	0.2	0.04	
Bpred	0.4	0.04	
Litter	2,680	335	
SOM	11,470	1,433.75	

It is common to apply a correction (“burn-in”) deleting part of the posterior, for example, the first half of the runs, to avoid the effect of the starting distribution (Gelman & Shirley, 2011). For this calibration, a sample of the last one hundred accepted parameter vectors was taken from the posterior distribution, and it was used for all further model runs, so every run was performed with 100 different parameter sets.

Input parameters of species

The KEYLINK model framework is conceptualized as an adaptable framework. Each user needs to determine for their specific site and questions the main drivers and pools required. Depending on the dataset, it is in general better to use less pools and equations if sparse data are available. Moreover, KEYLINK is not a soil fauna model and was not designed to simulate specific soil fauna species in detail. The soil fauna groups used consist of a wide range of species, for which average data are used. For a description of the species categories, we refer to the review on the KEYLINK concepts (Deckmyn et al., 2020).

Microbes and meso-macro fauna have a temperature curve using an optimum, minimum and maximum temperature. Each soil biota group also has its own maximum growth rate, C:N ratio, respiration rate and size. Death rate (d) is the inverse of turnover, mostly given in days. In Supplemental File S1 we briefly review main input parameters. We propose setting Ks, the concentration of the food source at which growth rate is half the maximum, equal to the existing concentrations for all meso- and macro fauna, so assuming growth could double at unlimited food source. But for microbial biomass the difference between growth of bacteria on a petri-dish unlimited in nutrients compared to field data of soil microbes clearly indicates that gmax in the soil is not comparable to laboratory data; if such data of gmax are used, the Ks should be increased considerably.

Calibration for Brasschaat pine forest

We show here the results from a calibration towards data measured and assumed, using proportions between fauna groups in Persson et al. (1980), in the Brasschaat Scots pine stand in Belgium. This forest stand is relatively well described in many publications concerning the trees and the total ecosystem fluxes, but less concerning the soil and very little was measured on soil fauna. Therefore, the partially assumed data refers to a hypothetical ecosystem that does not fully fit with reality in Brasschaat forest. We use this forest as an example of how the KEYLINK model can be used to improve our understanding of the system even when detailed soil faunal data are limited.

The parameters gmax, Ks and r are linked (increasing gmax has a similar effect to decreasing Ks or r). However, gmax or r ranges can be found in literature relatively easily. Therefore, we use fixed values for Ks (see Supplemental File S2) and parameterize gmax within the known limits. In this way, the number of parameters to be calibrated is 9, which is a reasonable number for most cases where limited data to calibrate towards are available. Of course, a user could decide to optimize more parameters if more data are available. A useful “rule of the thumb” is limiting the number of parameters to the square root of the number of calibration data available (Jörgensen, 2009), which means we can get a reasonable result for nine parameters assuming 81 data points.

In our case, no measurements of growth rates were available and information in the literature was scant. Therefore, we deliberately defined wide ranges for the prior values of each parameter to cover all the possible values found in the literature (Chuine, 2000; Linkosalo, Lappalainen & Hari, 2008). For species for which no prior parameter information was available, we assumed parameter values equal to the mean value of the range. The initial uncertainty of each parameter is quantified in terms of a prior probability distribution with lower and upper bounds. Because of lack of detailed knowledge, we assumed the distribution as uniform and non-correlated.

The gmax values were optimized using the prior range for gmax (Table 3). The data used to calibrate against were chosen to give a ‘standard’ procedure, so limited to biomass of the different C pools. Including all available data s.a. soil respiration, soil humidity could improve the run for Brasschaat, but would not be a representative run for the model. Other parameter settings, for example, sensitivity to C:N and recalcitrance, were based on model runs of the Brasschaat site by Deckmyn et al. (2011).

Table 3 Lower and upper bounds for the gmax prior probability distribution, for each one of the nine functional groups in the food web.

gmax	Lower bounds	Upper bounds	
Bacteria	1	3	
Fungi	0	3	
Mycorrhiza	1	3	
Bacterivores	0	2	
Fungivores	0	2	
Detritivores	0	0.5	
Engineers	0	0.5	
Herbivores	0	0.5	
Predators	0	0.5	

We ran the model for the time period 1999–2008, because this was the period in which the forest was still clearly dominated by Scots pine; since then a transition to more deciduous trees has been taking place. We calibrated towards stable C pools over the ten years for all C pools, with an allowed error margin of 20% for all faunal pools, except 10% for predators, and 12.5% for litter and SOM. Daily climate data (temperature and precipitation) were used. The full range of input data can be found in Supplemental File S2, except climate data, which can be downloaded with the model. Choosing to calibrate towards one or more pools can yield different results, and it depends on the end-user’s goal which calibration is preferred.

Model evaluation

Although coupling KEYLINK to real or simulated data of the aboveground ecosystem would yield more realistic results, in this exercise we used KEYLINK as a stand-alone model with quite constant input (e.g., litter, plant water uptake) to minimize the feedback effects and give a clear view on the model behaviour. This is a model evaluation, not a full model validation.

After calibration to the Brasschaat dataset, a set of scenarios was performed to evaluate the model: (I) Basic results; (II) Sensitivity to initial soil structure; (III) Changing initial litter C:N ratio; (IV) Changing initial litter recalcitrance; (V) Changing soil pH; (VI) Excluding predators.

Scenario I was done with the reference input parameters (Supplemental File S2), and used as a basal one to be compared with the other five alternative scenarios: scenario II with higher clay content in the soil (clay 15%); scenario III with lower litter C:N ratio (40); scenario IV with lower litter recalcitrance (20%); scenario V with higher pH (5.9); and scenario VI without predators by setting its initial biomass to 0 (Bpred = 0).

In each one of the five alternative scenarios, input parameters were the same than in the basal scenario, except for the parameter changed to generate the new scenario (see Supplemental File S2). All the six scenarios were run 100 times using a sample of 100 parameter vectors from the posterior distribution of the calibration, consisting each run in a simulation of 10 years at a daily time-step (3,653 days). Then, averages of biomasses were calculated for each C pool among the 100 simulations of 10 years, for each scenario, comparing the effects of disturbances on average values.

Results

Calibration

The model was run ca. 100,000 times with different parameter settings sampled from the prior parameter distribution. A sample of the posterior distribution was taken with the last 100 accepted parameter vectors for gmax (Table 4).

Table 4 Averages ± standard deviation from the sample of 100 gmax vectors from the posterior distribution of the KEYLINK model calibrated for the Brasschaat Scots pine forest.

	Sample gmax	
Bacteria	1.970 ± 0.424	
Fungi	0.295 ± 0.134	
Mycorrhiza	2.208 ± 0.302	
Bacterivores	0.205 ± 0.098	
Fungivores	0.095 ± 0.050	
Detritivores	0.091 ± 0.050	
Engineers	0.292 ± 0.062	
Herbivores	0.028 ± 0.018	
Predators	0.408 ± 0.060	

The optimization showed the link between the different groups of soil biota, for example, a high gmax for bacteria was coupled to a high gmax for bacterivores. The alternative five scenarios compared to the basal one can show very different results concerning specific C pools (Table 5). Running the model 100 times using the sample of the gmax values resulted in predictions with a quite wide range (Table 6).

Table 5 Effect of changes in input parameters on the average C pool (in g C m−3) size over 10 years.

Averages from 100 runs of ten years with gmax parameter sets of the sample from the posterior distribution. The “basal” column has the results using reference input parameters (Supplemental File S2), and the other columns show the results with lower litter recalcitrance (rec 20%), lower input litter C:N ratio (C:Nlit 40), higher pH (5.9), excluding predators (Bpred 0) and a higher clay content in the soil (clay 15%), respectively.

C pools	Basal	Rec 20%	CNlit 40	pH 5.9	Bpred 0	Clay 15%	
Bacteria (g C/m3)	5.98	6.94	6.35	5.90	2.67	4.75	
Fungi (g C/m3)	210.14	224.83	231.33	158.06	30.70	141.83	
Mycorrhiza (g C/m3)	39.83	40.20	40.03	38.70	29.75	39.27	
Bacterivores (g C/m3)	0.00	0.00	0.00	0.00	0.03	0.00	
Fungivores (g C/m3)	030	0.31	0.39	0.18	0.90	0.40	
Detritivores (g C/m3)	2.49	2.12	2.01	3.83	145.39	0.87	
Engineers (g C/m3)	0.04	0.04	0.04	0.51	1.54	0.05	
Herbivores (g C/m3)	0.12	0.12	0.13	0.02	5.48	0.16	
Predators (g C/m3)	3.33	2.86	2.87	6.41	0.00	1.36	
Litter (g C/m3)	3,695.08	3,262.71	3,481.79	3,728.85	2,727.81	4,245.54	
SOM (g C/m3)	10,825.04	10,941.01	10,742.97	9,347.23	3,175.08	11,655.70	

Table 6 Effect of changes in input parameters on the posterior distribution of C pool (in g C m−3) size over 10 years.

Minimum and maximum values within 100 runs of ten years with gmax parameter sets of the sample from the posterior distribution. The "basal" columns have the results using reference input parameters (Supplemental File S2), and the following columns show the same changes from basal as in Table 5. For C pool notation see Table 2.

C pools	Basal	Rec 20%	CNlit 40	pH 5.9	Bpred 0	Clay 15%	
(g C/m3)	Min	Max	Min	Max	Min	Max	Min	Max	Min	Max	Min	Max	
Bb	9.61E−42	670.64	3.41E−43	676.40	8.48E−38	683.87	0.00	669.74	0.00	547.97	8.06E−43	788.31	
Bf	2.63E−25	4,660.93	3.87E−25	4,587.16	3.74E−25	4,677.78	0.00	4,656.39	0.00	4,038.09	2.56E−25	4,763.21	
Bmyc	4.18	649.45	5.11	684.88	5.79	652.73	0.00	648.45	0.00	592.13	5.87	515.60	
Bbvores	6.89E−110	0.10	4.91E−105	0.10	1.17E−108	0.10	0.00	0.10	0.00	44.07	1.13E−96	0.10	
Bfvores	4.47E−84	194.08	5.82E−84	169.06	5.28E−83	204.50	0.00	141.68	0.00	351.94	1.27E−74	244.39	
Bdet	1.13E−67	1,720.93	3.76E−67	1,524.58	3.02E−68	1,567.73	0.00	6,898.50	0.00	5,997.90	7.56E−79	431.85	
Beng	7.13E−178	61.50	6.54E−176	33.87	5.45E−172	74.46	0.00	396.62	0,00	88.62	7.18E−140	43.86	
Bhvores	1.11E−68	49.65	5.23E−65	49.75	1.78E−65	49.64	3.78E−89	34.32	7.95E−22	85.00	1.12E−56	51.33	
Bpred	4.72E−21	1,616.28	4.73E−21	1,409.80	4.72E−21	1,466.09	0.00	5,512.93	0.00	0.28	4.60E−21	452.07	
Litter	6.0E+01	9,056.63	42.89	8,850.72	53.95	8,996.91	60.08	8,749.61	63.63	7,409.28	75.00	9,257.38	
SOM	2.35E+3	15,687.97	2,356.64	16,027.49	2,350.89	15,751.56	921.46	15,687.44	1,340.66	11,589.4	3,919.25	15,842.39	

Basic results

Mycorrhiza, herbivores and detritivores are relatively uncoupled, though influenced by predators, and follow the yearly climate curves. The bacterial and fungal biomasses are very strongly linked. The high gmax of bacteria allows steep peaks, which are generally followed by peaks in bacterivore biomass. As we used constant litter input into the soil and used a calculated constant fraction of potential evapotranspiration as water uptake from the soil, it cannot be expected that these results follow the normal annual trends in fluctuations of those fluxes. This can, at least partially, explain the relatively low bacterial biomass found in our results, since the bacteria would profit most from a rapidly changing environment, but under some unrealistic simulated conditions fungi could be displacing bacteria by competitive exclusion. For more realistic simulations the model can be coupled with other models that give that information as outputs, or with measured datasets.

All C pools tend to reach some stability after the first years, suggesting the model is well-balanced; however, stability values seem to be more sensitive to changes in gmax parameters for some pools (e.g., SOM). The set-up of the model, where we only calibrate the faunal gmax, does not allow calibration towards different ratio of litter and SOM decay. This depends on the uncalibrated parameter fragmentation, the sensitivity to recalcitrance, but also the temperature used for the litter and SOM. Here we used the same temperature, while in reality it would be expected that there would be certain differences in mean temperatures or their variability at different depths.

There was a high variability within the 100 simulations of each scenario, for example, basal scenario (Fig. 3), calling into question the reliability of predictions. We suggest here some theoretical predictions based on this example, but it is clear that for a realistic application it will be necessary to improve the calibration, by using more detailed data or by linking the model to a vegetation model. This highlights the relevance of developing databases including enough details for the key parameters of the different parts of the soil system (i.e., soil structure, hydrology, food webs).

Figure 3 C pools daily biomass averages and standard deviations from the basal scenario.

C pools (in g C m−3) averages (black) and standard deviations (grey) among 100 simulations of ten years using the gmax sample from the basal simulation scenario. (A) bacteria pool; (B) non-mycorrhizal fungi; (C) mycorrhizal fungi; (D) microbivores feeding on bacteria; (E) microbivores feeding on fungi; (F) non-engineer detritivores; (G) ecosystem engineers; (H) herbivores; (I) predators; (J) plant litter; (K) soil organic matter.

An overview of all C pools under the different simulation scenarios (Fig. 4) shows how changing one input parameter at a time influences the results. It must be taken into account that KEYLINK was run as a stand-alone model, which can explain why some of the resulting outputs seem not very realistic; linking it to a model or more detailed data of the aboveground ecosystem would greatly influence the results, but would not allow clear interpretation of the model functioning due to feedbacks. Since our goal here was to introduce the belowground model itself rather than a realistic application to a particular case, we chose to avoid those feedbacks with other parts of the ecosystem that are required for more realistic simulations. To further elucidate these effects and to show some of the potential outputs the model can give, we show a few of the most interesting fluxes (Table 7).

Figure 4 C pools daily biomass averages under different scenarios.

Averages of C pools (in g C m−3) among 100 simulations of ten years using the gmax sample, with the basal simulation (black solid lines), and the alternative scenarios: higher clay content (red dashed lines), lower input litter C:N (yellow dotted lines), excluding predators (dark blue dotted dashed lines), higher soil pH (green long dash lines) and lower litter recalcitrance (light blue, lines with two different dashes). Pools are the same as in Fig. 3, that is, litter (J), soil organic matter (SOM) (K), and the nine food web functional groups: bacteria (A), non-mycorrhizal fungi (B), mycorrhiza (C), bacterivores (D), fungivores (E), detritivores (F), engineers (G), herbivores (H) and predators (I).

Table 7 Effect of changes in input parameter on major output fluxes over 10 years.

The first three rows show bacterial, fungal and mycorrhiza respiration (R) fluxes (g C m−3), respectively. The next three rows show the total turnover (g C m−3) on an organic matter pool carried out by bacteria (Bact) or engineers (Eng) over 10 years. The penultimate row shows the fungi to bacteria ratio. The last row is soil water content (SWC, l m−3). Columns show average values and standard deviations from 100 runs of ten years from the sample of the posterior distribution, with a basal scenario using reference input parameters (Supplemental File S2), and the same changes from it as in Table 5.

	Units	Basal	Rec 20%	C:Nlit 40	pH 5.9	Bpred 0	Clay 15%	
Rbact	g C m−3	250.2 ± 222.8	293.6 ± 227.2	262.3 ± 225.7	248.9 ± 222.4	119.3 ± 149.9	204.5 ± 225.9	
Rfun	g C m−3	3,924.8 ± 5,043.6	4,190.0 ± 5,035.6	4,215.4 ± 5,166.1	3,400.6 ± 4,776.3	963.6 ± 2,669.4	2,638.5 ± 4,270.7	
Rmyc	g C m−3	773.9 ± 443.7	785.3 ± 447.3	779.1 ± 445.9	760.4 ± 431.3	560.3 ± 335.5	760.5 ± 447.5	
Bact SOM turnover	g C m−3	995.6 ± 758.2	1,100.2 ± 716.3	1,036.5 ± 757.7	985.4 ± 752.6	438.6 ± 524.2	780.2 ± 741.7	
Bact litter turnover	g C m−3	258.6 ± 264.1	367.7 ± 328.6	280.2 ± 276.2	257.6 ± 263.9	119.6 ± 170.6	223.5 ± 279.7	
Eng litter turnover	g C m−3	1.5 ± 5.3	1.3 ± 4.5	1.3 ± 4.4	43.8 ± 104.4	97.3 ± 72.1	2.0 ± 5.8	
Bfungi/Bbact	–	35.2	32.4	36.4	26.8	11.5	29.9	
SWC	l m−3	147.9 ± 32.7	149.2 ± 32.4	150.3 ± 33.7	144.2 ± 31.1	134.2 ± 21.8	321.6 ± 27.2	

The simulated scenarios showed that increasing clay content (i.e., changing initial soil structure) resulted in an increase in water content (Fig. 5) and a decrease in litter and SOM decay while fungal/bacterial ratio decreased. In fact, this scenario caused the largest change in soil water content, showing the sensitivity of the system to initial soil structure and its crucial role for soil hydrology. On the other hand, lowering litter recalcitrance or C:N ratio resulted in an increase in microbial biomass, mainly fungi, which caused an increase in litter decay, while SOM decay did not show a clear change. The scenario with higher pH allowed engineer species (and predators with them) to increase in biomass, altering the soil structure with an increase in macroporosity, which caused a clear decrease in SOM stabilization. Finally, the exclusion of predators totally changed the soil porosity and the trophic interactions along the food web, causing the largest increase in the decay of SOM and litter, which highlights the crucial role of predators in the regulation of the soil C cycle.

Figure 5 Daily volume averages of soil water content (SWC) and pore size classes in the soil matrix.

Means of volume (in l m−3) among 100 simulations of ten years using the sample of gmax vectors, for the evaluation scenarios (see Fig. 4). Graphs (A)–(D) for pore size classes, (E) for SWC. The inaccessible pore size class is not shown because it was not affected by changes in porosity.

Discussion

The Brasschaat forest is sandy, with low pH and recalcitrant litter; as expected, this is an environment not suited to earthworms. The model correctly simulated extremely low values of engineer biomass. Increasing the pH increased the engineers pool, for example, earthworms population, but this remained too low to have a significant impact on the system. This is quite realistic as neither litter quality nor soil quality are ideal for earthworms. Obviously, to calibrate the specific parameters concerning earthworms the Brasschaat forest is not an ideal site.

The high variability observed in some populations could make them appear more unstable than what can be expected in reality, and it is indeed expectable that a more realistic application of the model will yield different results. However, there are also empirical evidences that some populations, for example, microbial biomasses, can be very unstable depending on hydrology (Blackwell et al., 2010; Zhao et al., 2010), showing short-term spikes in biomass as those observed in the presented simulations, in response to predicted changes in water availability.

Changes in litter quality (i.e., in C:N ratio and recalcitrance) caused small differences between scenarios, being the simulations from those two alternative scenarios quite similar to the basal scenario. Despite the apparent increase in litter decay for lower litter recalcitrance and for lower C:N ratio, as mentioned before, the high variability among simulations (Fig. 3) suggests that differences between scenarios caused by changes in litter quality are negligible. It is well known that litter quality is one of the main factors controlling litter decomposition rates (Zhang et al., 2008; García-Palacios et al., 2013), and the observed trend fits with experimental evidence, but clearer effects of changes in litter quality are expected. Therefore, these simulations should be tested again, particularly with the model linked to a vegetation model, and, if necessary, improvements should be added.

Changes in textural porosity, on the other hand, showed clearer effects on soil processes. The scenario changing clay content to increase textural porosity caused the highest impact on hydrology, increasing soil water content (Fig. 5E), which led to a decrease in SOM availability, decreasing microbial community biomasses and, therefore, decreasing the decomposition of SOM and litter. However, changes in C pools were not as clear as in hydrology, mainly due to the high uncertainty in calibration results.

The scenario without predators showed the most interesting results because the interactions between the different food web parts are apparent, and it showed a high contrast with the basal scenario. Exclusion of predators, setting the starting biomass of that pool at 0, showed how the model tracks its crucial role in the ecosystem (Fig. 4). Predators produce a top-down trophic cascade on the food web, for example, on herbivores and roots. Microbial decay is reduced as fungi and bacteria are consumed by the increased populations of bacterivores and fungivores. Despite of the decrease in bacteria and fungi, SOM and litter were also lower without predators. This could be explained by an increase in engineer populations with implications also on the soil matrix. Overall the model successfully tracked soil food web dynamics and also their interactions with soil porosity. The effect of larger soil predators (e.g., Araneae, Carabidae, Formicidae) slowing down SOM decomposition and enhancing its stabilization has been previously found in experiments (Kajak, 1995), as well as mycorrhiza effect on porosity by making aggregates (Siddiky et al., 2012). Those trophic cascade effects over SOM stabilization depends on environmental conditions such as rainfall, with predation on microbivores reducing litter decomposition rates in more humid sites, while reductions in rainfall could lead to a shift in that trend with predation on microbivores indirectly increasing litter decomposition rates (Lensing & Wise, 2006). The predicted effects of predator exclusion increasing SOM and litter decay have been found in different ecosystems, for example, grasslands (Kajak, 1997) and forests (Lawrence & Wise, 2000), but contrasting results (e.g., Kajak, 1995; Lawrence & Wise, 2004) suggest that it is not a general pattern. Considering the high rainfall conditions at the modelled Belgian forest, and according to the suggested trends based on experimental research, soil predators feeding on microbivores at Brasschaat forest could promote SOM stabilization, which would fit with the simulated scenarios. Therefore, KEYLINK model seems to fit with the expected food web and C dynamics, and could serve to improve the biogeochemical cycles modelling, as is needed for larger scale predictions (Grandy et al., 2016), by coupling it with other ecosystem models.

Hydrology is influenced by aggregation and by macropore formation by ecosystem engineers. The increased macroporosity increases infiltration rate with reduced water-logging and runoff. Predators have a clear indirect effect on soil porosity by consuming engineer species, and also microbivore species, which leads to changes in soil hydrology (Fig. 5). Variations in bacterial pore and micropore volume are positively correlated, while mesopore variations are negatively correlated with both; the higher volume in mesopores, the lower in the two other classes, and the faster the water drains from the soil layer. That is what we can expect to happen in real soils, so the model seems to simulate appropriately those dynamics. The increase in macro and mesoporosity volumes without predators, so with higher engineers, resulted in a decrease of soil water content of 9.26% (increasing the pore aeration), and under those conditions the availability of SOM and litter for bacteria and fungi could be increased, explaining why SOM and litter are lower even with lower bacteria and fungi. This highlights the role of hydrology on trophic cascade processes, which can be enhanced or reduced by water distribution through the soil matrix (Erktan, Or & Scheu, 2020), and also the relevance of considering how climate change effects on soil structure, hydrology and food web interactions (particularly trophic cascades) can affect microbial communities (Thakur & Geisen, 2019) and, therefore, litter and SOM decomposition.

The aim of this study was to present a first version of a new concept model that hopefully will serve to challenge current state-of-the-art soil modelling. But we are aware that to do that we will need to improve the calibration of the model in the future, using more complete databases that take into account all the elements needed to calibrate KEYLINK, which, on the other hand, are currently extremely scarce. By presenting this concept model that challenges the current way of simulating soil biochemical cycling, we hope to stimulate that future studies will also be designed to take into account the pools and functional groups needed to calibrate KEYLINK.

Conclusions

KEYLINK is a relatively simple, fast and easy to modify soil model that can be used as a stand-alone model to understand soil systems, or linked to detailed aboveground data/models to predict SOM turnover. Model evaluation showed that KEYLINK is capable of simulating properly not only the soil food web and C pools dynamics, but also how they interact with soil porosity and hydrology, which is one of the main goals of this new model. The results from the evaluation scenarios showed that SOM turnover is driven not only by microbial biomass, but also by soil structure and hydrology. Moreover, microbial biomass is strongly regulated by the presence/absence of the other soil fauna. Especially the effects of the predators and the ecosystem engineers are extremely significant for our understanding of soil functioning. Furthermore, since management can differentially affect the larger soil fauna, KEYLINK can be of great use to investigate potential effects of management changes on soil SOM, nutrient turnover and hydrology.

This model shows degradability of SOM can be adequately simulated from accessibility in relation to pore space instead of the existing concepts of slow and fast pools. This allows a closer link to the soil structure and soil fauna which we consider closer to the actual, and follows the concepts as first described by Kuka, Franko & Rühlmann (2007), but applied here in a wider framework and including the hydrology.

For a full validation or better calibration of the model, datasets are required including basic data on the aboveground, for example, litter input, water uptake, root growth and turnover, in combination with relatively detailed data on soil structure, that is, pore size distribution, and hydrology and soil biota, for example, biomass of bacteria, fungi, mycorrhizal fungi and main meso-and macrofauna. All these data are available, but very seldom at one site as most studies are focused on one or other aspect of soil science.

In conclusion, KEYLINK is a step towards a new generation of ecosystems models that include functional diversity, trophic structures and ecological processes as important factors shaping soil/ecosystem carbon and water cycling. Future versions, fed by more detailed data, will need to be developed in order to improve our current predictive capacity.

Supplemental Information

Supplemental Information 1 Review of input parameters and carbon pools.

Click here for additional data file.

Supplemental Information 2 Input parameters for Brasschaat run.

Click here for additional data file.

Supplemental Information 3 Precipitation and temperature in Brasschaat forest during 10 years.

Click here for additional data file.

The authors express their gratitude to all the people who have contributed to the BioLink and KEYSOM COST Actions, whose work contributed also to the development of the KEYLINK model.

Additional Information and Declarations

Competing Interests

Author Contributions

Data Availability

Jorge Curiel Yuste is an Academic Editor for PeerJ. Mathieu Javaux, Harry Vereecken and Andrea Schnepf are employed by Forschungszentrum Jülich GmbH.

Omar Flores conceived and designed the experiments, performed the experiments, analyzed the data, prepared figures and/or tables, authored or reviewed drafts of the paper, and approved the final draft.

Gaby Deckmyn conceived and designed the experiments, performed the experiments, analyzed the data, prepared figures and/or tables, authored or reviewed drafts of the paper, and approved the final draft.

Jorge Curiel Yuste conceived and designed the experiments, analyzed the data, prepared figures and/or tables, authored or reviewed drafts of the paper, and approved the final draft.

Mathieu Javaux conceived and designed the experiments, authored or reviewed drafts of the paper, and approved the final draft.

Alexei Uvarov conceived and designed the experiments, authored or reviewed drafts of the paper, and approved the final draft.

Sietse van der Linde conceived and designed the experiments, authored or reviewed drafts of the paper, and approved the final draft.

Bruno De Vos conceived and designed the experiments, authored or reviewed drafts of the paper, and approved the final draft.

Harry Vereecken conceived and designed the experiments, authored or reviewed drafts of the paper, and approved the final draft.

Juan Jiménez conceived and designed the experiments, authored or reviewed drafts of the paper, and approved the final draft.

Olga Vinduskova analyzed the data, prepared figures and/or tables, authored or reviewed drafts of the paper, and approved the final draft.

Andrea Schnepf conceived and designed the experiments, performed the experiments, authored or reviewed drafts of the paper, and approved the final draft.

The following information was supplied regarding data availability:

Codes are available at GitHub: https://github.com/Plant-Root-Soil-Interactions-Modelling/KEYLINK.

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
