# Peer review of "KEYLINK: towards a more integrative soil representation for inclusion in ecosystem scale models—II: model description, implementation and testing"

_PeerJ, doi:10.7717/peerj.10707_

## Round 0.1 · original submission · Major Revisions

Three independent reviews have been received of which two reviews are positive and one negative. I believe that the authors deserve a chance to respond to these questions, and therefore propose major revisions. In addition to the many questions raised by the reviewers, I think the authors carefully need to consider Reviewer 1´s comments on the existing literature on the dynamics of soil macro-fauna and dimension errors. Reviewer 2 also highlights an important issue with the dependency to the accompanying manuscript “review and model concept”, which will carefully be monitored. Despite knowing the final fate of the review and model concept manuscript, this manuscript needs an improved background section to be able to have a chance to stand alone.

·

Basic reporting

I review lots of manuscripts with models, but when I meet dimension errors, I cease further reviewing. Unfortunately this applies to the current ms

Experimental design

Acceptable, but not great. A lot of literature exists on the dynamics of soil macro-fauna and the authors only mention that they found little information.

Validity of the findings

NA

Additional comments

General: What I really like in Keylink is the attempt to integrate phsyical, chemical and biological aspects of the soil.
I do think that the result can potentially be valuable for management purposes and assessments.

When I see the term "model" as theoretical biologist, I think of a list motivated assumptions from which a mathematical model follows,
which properties can be studied and realism tested. To be honest, it was a bit of a disappointment to see that the paper is
in fact a description of computer code, and the result even depends on the sequence in which the various steps are calculated.
The many if-then statements make it impossible to study the mathematical properties. So it will be of little help to gain more
(theoretical) insight in soil dynamics, but can, nonetheless, still be usefull in practical applications.
I do understand that the type of mathematical model that had in mind can easily be too simple to match reality close enough,
but the implication is that validation will be very hard, which the authors in fact admit.

Living in a country where N-deposition is a key issue for soil pH, I was missing how this is incorporated in the model.

What was not really clear to me is the time-scale the authors had in mind in the background of seasonal forcing.
Apart from cycles in temperature and precipitation, leaf litter is of course dominating food availability for soil macro-fauna.
Earthworms easily life for 7 years (depending on temperature); the simulated period 1999-2008 is in fact matching this life span.
My complaint boils down to the notion that the memory of the various player of the game, from bacteria to macro-fauna differs by
order of magnitude, which I do not see reflected in the structure of the model.
The only memory that the model accounts for is the existing pools.

Line 216 introduces growth G, but G is not specified. Previously g_max was introduced, with dim(g_max) = #/time, which follows from
d/dt N = g_max.
From (10) follows that the units of R is g C/d.m^3 and not g C/m^3, like the line above (1) suggest.
In (27), however, g_max has the units of R, which amounts as a dimension error.
In (28) and (29), g_max has the units of CN_eng, and even time disappeared from the units of g_max.
There are more issues with units and dimensions.
In (39), DOM is in g/m^3, f_DOM is a fraction, but respiration R is a rate.

When I notice dimension issue, I cease further reviewing. Sorry for this.

·

Basic reporting

As was stated by the authors in their letter to the editor, this is a follow up paper on model tat is described in an unpublished paper. I have no problem with this, more how it's referenced:
(G Deckmyn et al., 2020, unpublished data)
I think it should be:
(G Deckmyn et al., in review)
There are 2 problems with the reference, you have put a year on it, and you have choosen the word data. To me, that indicates that there is data that support a statement, but it hasn't been published yet. But that's not the case here.

In any case, this manuscript is not publishable without the first paper, so I guess that will sort it self.

A part from this, I think (given the circumstances I can not be sure) that the background is sufficent. In all sections but the model description this was the case. And this is maybe something that needs to be changed, even though you have the accompanying paper, there are no references to the logic/science behind using certain formulations. The current setup makes it impossible to follow witout having the other paper.

A comment on the notation CN ratio, I think it should either be C-N ratio or C-N ratio (C:N) the first time, and then C:N throughout the text. And also on the same, C is defined in the Abstract. The Abstract is a stand alone part of the manuscript, any definitions made there have to be redefined in the main text. N is never defined.

Experimental design

no comment

Validity of the findings

no comment

Additional comments

A part from the comments I have made above, I think it is a good manuscript, it was easy to read and how the assumptions that have been made in both model and experimental design affected the results were clear and easy to follow.

Even with accompanying paper, I think some background is needed in the model description part of the paper.

Reviewer 3 ·

Basic reporting

'no comment'

Experimental design

'no comment'

Validity of the findings

'no comment'

Additional comments

I read the manuscript by Flores et al. about an ecosystem model (KEYLINK) to integrate soil biological and physical processes to predict soil processes like carbon dynamics. In general, I like the modeling approach, and the details provided by the authors. I, however, have little background on soil hydrology and I am not an ecosystem modeler myself. My background is more soil biology and particularly soil food webs. Authors will realize that when reading some of my concerns below and probability some of my naivety on the topic.

Line 51: remove “shaping both”.
Line 57: perhaps use Carbon (C) for the first time, and then use C throughout the manuscript.
Line 63: soil organic matter (SOM).
Line 64: this sentence reads odd. Do you mean “the relevance of other processes in the physical protection…”
Lines 67: do these models have full name? If so, please provide them before abbreviating!
Line 68: ecosystem engineers.
Line 72: what are detailed soil models? Not clear!
Line 75: I again wonder if there is a full form of KEYLINK? If not, why is it called KEYLINK?
Line 77: such as? Provide some examples what those parameters are in extant models?
Line 87: please use ecosystem engineers throughout the manuscript.
Lines 81-88: were there also different thermal and moisture sensitivity modeled for soil organisms? For example, large body sized soil organisms could be more sensitive to higher temperature.
Lines 89-90: I understand that authors have another paper with more scientific background about their approach. It, however, is unpublished. So as reviewer, I have no access to it to understand the scientific background behind the model. It will be therefore a favor to the readers if authors provide at least with few sentences about the scientific background of the model.
Line 118: which ecosystem engineers authors have in mind here? Earthworms? Adult earthworms? The size differences between adult and juvenile ecosystem engineers may have differences in accessibility to soil pores.
Lines 129-130: are these densities general? Won’t they differ with soil type?
Line 198: did not get why authors have inverted comma for found?
Line 200: does completely dry pores have no access to a consumer?
Lines 633: not sure why soil bacteria would profit most from a rapidly changing environment. Please clarify.
Lines 651-656: there could be some other small sized ecosystem engineers, such as enchytraeids. Did authors consider “other ecosystem engineers” of the soil than earthworms?
Lines 659-660: which engineers increased? Do engineers then compensate for decay reduction? Also, some ecosystem engineers stimulate bacterial growth. How does that balance out?
Line 666: the loss of predator effects was exclusively owing to large predators or all body sized predators. Please make this clear as this is so far the most interesting model outcome.
Line 687: are authors developing any software to make KEYLINK accessible to ecosystem modelers?

---

## Round 0.2 · Major Revisions

The resubmission has now been assessed by three reviewers, one of whom one is new (reviewer 4) since only two of the three original reviewers assign for a second review. Reviewer 4 raises a lot of concern regarding the parameterization and of the interpretation of the calibration. Based on the comments by Reviewer 4 I recommend a new major revision of the manuscript.

·

Basic reporting

No comment

Experimental design

No comment

Validity of the findings

No comment

Additional comments

Overall, the authors updates really improved the manuscript.

There is only one thing, which I missed in my previous assessment.

"... soil structure as ecosystem engineers are predated...", this is the first mentioning of the term ecosystem engineer.
From ecology, ecosystem engineers are organisms that creates, significantly modifies or maintains an ecosystem.
Sure earthworms are doing that, but so are beavers, sphagnum mosses, kelp and some bacteria.
It would be much better to replace ecosystem engineer with earth worms throughout the document, since that is what you are referring to. But of course acknowledging that earth worms can be considered to be ecosystem engineers.
One reason I am assuming that is because you are allowing pH to directly affect the group, not sure that ants would be as affected by pH as earth worms are.
If not, then it has to be very clear that you in this study how you are defining ecosystem engineers.

Reviewer 3 ·

Basic reporting

Discussion needs more flesh.

Experimental design

No comment.

Validity of the findings

No comment.

Additional comments

I thank Flores et al for revising their manuscript and addressing my concerns in their revised manuscript. The revised manuscript has improved, and it complements well with their other review manuscript (Deckmyn et al.). I have mainly suggestions to improve their discussion a bit. For instance, they have three citations in total in their entire discussion. Authors should discuss their results with a little more context before the manuscript is accepted. For example, there were two recent review papers out highlighting the importance of soil predators in driving soil microbiome (Thakur and Geisen 2019, Trends in Microbiology and Erktan et al. 2020, Soil Biology & Biochemistry), which could easily be integrated in their discussion (lines 668-682). Another minor issue is the way authors claim KELYLINK as “one of the first and most ambitious attempts ….”. I will be careful and tone it down-particularly “the most ambitious attempts”. The authors should leave that to the readers to judge!

Reviewer 4 ·

Basic reporting

Language use throughout the article could use improvement. There are numerous grammatical issues throughout the text and the writing style at some points is less formal than typical for academic articles. I would recommend thorough proofreading of the text for grammar and style. In addition, at some points the text reads less as a scientific description and evaluation of the model and more as an advertisement for others to use the model.

Literature references are adequate, taking into account that the scientific context for the model, including a more extensive literature review, was mostly presented in a separate companion paper. There are a few specific model structural choices that could use more explanation or citation support.

Experimental design

The general structure of the model makes good sense in terms of process representation and the goals of the model. The integration of biological factors and their impacts on soil porosity and structure are interesting and well based in soil science. There are some issues with equations and organization of the model description, but overall I think the model design is sound.

The application of the model and calibration of parameters, however, have a lot of problems and I think the model simulations overall fail to reach accepted technical standards. The model has a large number of pools (13) and a lot of parameters that are poorly constrained by site measurements. Parameter estimation was conducted using a Bayesian Markov Change Monte Carlo approach using only 9 data points, of which several were literature-based estimates rather than site estimates. The study treats these 9 data points (representing a biomass number for each of 9 pools) as 99 points by assuming steady state across years, which I think is unjustifiable pseudo-replication. Overall, I think a Bayesian calibration of a model with 13 C pools using only a total of 9 data points cannot be justified. This is borne out in the results, such as Tables 5 and 6 which show huge ranges in parameter value distributions. In addition, mean pool values (Table 5) are not at all consistent with the data values (Table 2), suggesting that the calibration was unsuccessful. For example, fungal biomass was calibrated to a value of 15 g C m-3 (Table 2) but in model simulations has a mean value of 200 (Table 5).

It does not seem that there is enough data available for the study’s field site to actually constrain the parameters of the model, so it’s hard for me to say what the best path forward might be. I think that for this study to be successful the calibration procedure needs to be completely re-conceived and rewritten in a way that is consistent with the availability of data to constrain the model’s parameters. It is extremely important to evaluate the calibration procedure itself, which clearly failed in the results presented. Careful examination of posterior parameter distributions and the pattern of the MCMC results is important for establishing whether the parameterization procedure actually converged in a useful way.

Perhaps a more theoretical analysis of the model behavior, using reasonable parameter values and focusing on understanding interactions among different pools in more detail, would be one path forward. Overall I think this model is a good example both of the process insights that can be gained by increasing model complexity and an example of the difficulty of getting meaningful results from an increasingly complex model with parameters that are difficult to constrain.

One of the most promising aspects of the model is the connection between faunal engineers and soil structural changes. It seems like a missed opportunity to run the model only at a site with very low faunal engineer biomass, where this effect cannot be investigated fully.

Validity of the findings

The results section of the manuscript is very problematic. The poor design of the model calibration and the complexity of the model itself produced results that are difficult to interpret and inconsistent with reality. The interpretations of the results in the manuscript take an optimistic view that is not consistent with the actual results. Model simulations produced a very wide range of outcomes as evident in Table 5. Biomass time series (Figure 3) look very unstable and are characterized by short-term spikes in biomass of some pools, including huge variability in bacterial biomass, that is not realistic for an ecosystem-scale soil model. Because only the mean of multiple simulations is shown, it is difficult to tell how these time series varied among simulations (within the same scenario) but I expect the variability is very high, calling into question many of the interpretations of the results. It would help to see the variability in simulations behind Fig 3. I would not be surprised if individual simulations within each scenario were extremely variable due to parameter uncertainty (consistent with Table 5 and 6).

The analysis of the results is very short and quite shallow and does not address the key aspects of the different scenarios that were simulated. There is not really a meaningful analysis of the model simulations beyond a cursory, qualitative description. The results and discussion seem to start from the assumption that the model will be useful because of the processes it includes, and do not reflect an actual objective analysis of the simulation results. There is a lot of complexity in the model results which is mostly overlooked. I think to meet scientific standards the results and discussion would need to be completely rewritten to reflect an objective assessment of the model results.

Overall, the results and discussion are more an argument trying to justify the value of the model’s structure than a real analysis of the model outcomes. The analysis of the calibration procedure does not take a serious critical view of the possibility that the Bayesian approach might not give strong constraints on the model parameters. Part of the value of a Bayesian analysis is that it provides estimates of parameter uncertainties, and insights about whether parameters could be well constrained by available data, and it seems that the analysis did not take advantage of this but rather started with the assumption that the model could be well constrained despite the limited measurements that were used.

Additional comments

Line 31: I suggest starting the abstract with a sentence about the scientific context or knowledge gap that motivate the model before jumping into the model description

Lines 34-37: This reads as more of an advertisement for the model than a statement about the science. In this study, KEYLINK was not coupled to another ecosystem model so it does not seem relevant to the study to say how it could be coupled.

Line 50: The model was not actually compared with a first-order model, so there isn’t evidence that it was actually a more successful approach.

Line 59: Century and RothC are not the oldest soil models that exist. It would be more accurate to just say that they are widely used

Line 102: I would start the methodology section with an overview of the whole model and how the pieces relate to each other as shown in Fig 1.

Line 114: If earthworms eat all soil, do they have access to all pore sizes, or only the bacterial pores? It is not clear from the description

Line 127-129: The statement about macroporosity should have a citation to literature supporting it. And this statement does not seem to fit with this section since none of the other pore size classes are described in terms of laboratory measurements.

Line 139: Equation 1 should have a + between the terms in the denominator, not a -

Line 153-155: Is evaporation assumed to be equal to potential evapotranspiration? This does not seem realistic since it ignores source limitation of evaporation as well as boundary layer and conductance effects that limit actual evapotranspiration

Line 199: Equation 9 should include terms for modifiers to gmax (temperature, pH, etc). The sum notation should indicate the index that is being summed over. The role of excreted faeces also needs to be in this equation, otherwise it states that all of the substrate is being converted to biomass growth which is untrue.

Line 205-207: An equation should be shown or referenced for fa

Line 208: Physical recalcitrance is far from a novel concept, and has been included in conceptual and numerical models prior to KEYLINK (e.g., MIMICS, and the passive pool in CENTURY)

Line 218: The model simulates increases in biomass of pools, not in population number

Line 225: It’s confusing to refer to temperature sensitivity in growth when gmax as presented so far does not include temperature dependence (Eq. 9). Adding these factors to Eq. 9 would make this clearer.

Line 235: This equation states that predation rate depends only on the predator’s total growth rate from all substrates, and not on the biomass of the prey. This does not make sense. If a predator feeds on multiple types of prey, then this suggests that all are predated at the same rate regardless of their different biomass amounts. This equation would make sense if it refers to the fraction of a predator’s growth from a single prey type, but that is not what the equation actually shows.

Line 248: availability was f_a in Eq. 9, but is noted as (a) here

Line 254-312: This description would be easier to follow if it was moved to be right after Eq. 9 (which describes gmax), or if there were a statement after Eq. 9 stating that modifiers to gmax were described below. Either way, Eq. 9 should show that gmax is modified by additional factors.

Line 259: Density-dependent microbial turnover should be a modifier on microbial death rate, not on growth rate

Line 271: This equation has a discontinuity in it at T=Tmax, where mT goes directly from 1 to 0. This doesn’t make much biological sense

Line 280-281: The text should reference literature supporting the pH effects on bacteria and fungi

Line 283-284: These equations do not seem correct. gmax would approach infinity as pH approaches the threshold in each case. For example, at pH=8.01, mpH for fungi would be 10. An exponential function would work better. Also, Eq. 18 defines a declining fungal growth at high pH and constant fungal growth at lower pH (< 8), while the text above refers to an increasing fungal growth at low pH.

Line 287: It is not clear why there is a linear response of engineer saprotrophs to pH but a 1/pH response for bacteria and fungi. Is there some literature support for this choice?

Line 291-293: This sentence does not make grammatical sense and needs to be rewritten.

Line 294: mrec should be defined when it is introduced

Line 303: Why was a linear equation chosen here and a power law above?

Line 314: This section is not really about closing the C budget, it is about the fraction of decomposed material that is converted to faeces (which also needs to be included in Eq. 9)

Line 325 and 328: Recalcitrance is not a conserved mass quantity and does not have a budget

Line 335: Based on the text, N limitation should be modifying r rather than gmax

Line 336: This equation states that growth rate is positively correlated with C:N of SOM, the opposite of the statement here that growth slows with lower N

Line 348-349: What specific parameter is “twice as recalcitrant” referring to?

Line 360-361: These units do not make sense.

Line 363-364: Eq. 31 burrow volume as being directly proportional to engineer biomass, in an instantaneous way. It does not make sense to pair this with a turnover rate. If rates are defined, then burrow volume needs to have both a formation and a turnover rate.

Line 369: Should these units be m3/m3?

Line 395: These units are also incorrect

Line 398: Can layer thickness change in the model? Based on other equations, it does not seem like it, so this statement is confusing

Line 423-446: This section does not actually specify how much DOM is leached

Line 472-474: Since KEYLINK has not actually been modified or calibrated for different ecosystems or coupled to any other ecosystem models, this statement is unsupported. I think this whole paragraph (except for the Github link) could be removed since it is mostly an advertisement of the model and not a scientific statement.

Line 480-481: This text should specify why multiple runs are needed (to explore parameter space). generally, I think this would be specific to the uncertainty in model parameters rather than general to the model itself, so I’m not sure it makes sense as a general recommendation.

Lines 506-531: I think these lines could be removed. It is not necessary to describe the basics of how Bayesian parameter estimation works. Previous literature could be cited instead.

Line 524-525: This is the only mention of a drylands version of the model, and seems irrelevant to the rest of the manuscript.

Line 536: Assuming that 9 data points are equivalent to 99 data points by assuming pool values are constant is unjustifiable. I think this is a fatal flaw of the parameterization approach.

Line 550-552: This is not specific to this study and could be removed.

Line 575: R is respiration rate predicted by the model. It is not a parameter. Should this be r?

Line 578: This calibration procedure used 9 data points to constrain 9 model parameters. There is much too little data for this to be a workable approach.

Line 582: There are not 81 data points. There are 9 data points.

Line 584: “no measurements were available” — measurements of what exaclty?

Lines 614-617: Clay, litter recalcitrance, and litter C:N are not included in Table 1

Line 621: What is LHS?

Line 606-624: Was the model calibrated to steady state for each scenario? Or did it use the baseline parameters for all scenarios? If the second is true, then the model was likely out of steady state for other scenarios, making the results unreliable

Line 630-634: Based on Table 5, it looks like the model parameters were not well constrained at all. The +/- values don't make much sense as they include negative biomass for most pools. I imagine the distributions are skewed, so it would make more sense to show a figure with actual distributions rather than report a standard deviation that is not a good measure of actual variance. Showing posterior parameter distributions would provide better information about whether the MCMC approach was actually effective at constraining parameters.

Line 635: What is meant by “relatively uncoupled?” They were poorly correlated with each other?

Line 637-638: The behavior of bacteria in the model is clearly unrealistic, with a huge biomass spike at the beginning of the simulation followed by death of basically the entire bacterial community.

Line 642: “bacteria would profit most from a rapidly changing environment” This doesn’t make much sense either in the model or in real life. Typical soils have large bacterial populations whether they are rapidly changing or not

Line 645: Looking at Figure 3, the opposite of this statement is true. C pools seem very unstable and are characterized by spikes and large fluctuations which suggests to me that the model is poorly balanced. One scenario lost 75% of SOM in the first year!

Line 651: Different soil layers should have the same long-term average temperature, although shallower soil layers would be expected to have wider fluctuations

Line 658-659: The study included six different model scenarios, but only one is discussed and only in one sentence here.

Figure 2: The diagram should show which pools are external to the model (tree shoots, litter, CO2) and which are actually model pools. From the text, it seems like the SOM pools shown (different POM sizes, DOC) are not actually in the model, so it is misleading to show these as model components in the diagram. The diagram should be consistent with the model that was actually used in the study.

The colors of the lines in Figures 3 and 4 are very difficult to tell apart.

---

## Round 0.3 · accepted · Accept

I agree with reviewer 4 that the manuscript has improved and gladly suggest it to be accepted. I also encourage the authors to share the code used for Bayesian calibration together with the model output or code to recreate the actual full set of simulations, which also reviewer 4 suggests.

Reviewer 4 ·

Basic reporting

The style of writing in the manuscript has been improved and the issues with clarity and English use have been mostly addressed. Literature references and background provide good context for the modeling study. The article structure and organization are good, and figures and tables are clear and work well.

In terms of raw data sharing: The model code is available on Github and the meteorological drivers are included in supplementary material. However, the code used for Bayesian calibration is not posted, and the model output or code to recreate the actual full set of simulations is not made available. For complete data sharing, I would suggest adding the code used for calibration to the Github repository and posting the model output and/or a script for running the actual simulations that were analyzed in the results.

Experimental design

The design of the model is sound. The calibration procedure is not very robust due to lack of data, as I noted in my previous review. I do think that the revised framing of the calibration procedure and interpretation of the results has mitigated most of these issues. The simulations are now explained more clearly as an initial test of the model, and the results are not interpreted as being quantitatively predictive. I think the Bayesian calibration procedure does make sense when viewed as a strategy for balancing the different model parameters to produce a simulation that is relatively close to steady state for the measured pools, for the purpose of investigating the model’s behavior, and the paper now acknowledges that this calibration procedure should not be viewed as a strong constraint on actual model parameters or quantitative predictions. Overall I think this framing addresses the main issues with the calibration procedure that I brought up in my previous review.

Validity of the findings

I think the manuscript has been improved in terms of the framing of the conclusions and results. In terms of underlying data, as I noted above I would suggest providing scripts and/or model output data for the calibration procedure and the actual simulations that were analyzed in the manuscript.

Overall I think the revised manuscript gives a good overview of the model and demonstration of how it works, including both strengths and weaknesses of the model. There are certainly areas where the model and its application could be improved in the future, but I think the revised manuscript does a better job of acknowledging the weaker areas of this initial model application and sets a baseline for future improvements.

---

## Author Rebuttal · Round 0.3

November 16th, 2020

Dear Editor,

We thank the reviewers for their effort reviewing our work, and for their comments that have helped us to improve our manuscript. We have edited it following their advices, and we provide in this rebuttal letter the requested answers to address their concerns. We have also modified and replaced figures 2, 3, 4 and 5 (being figure 3 a new one requested by reviewer 4), tables 2 and 5, a new table 6 (being the former 6 now the table 7), and supplemental file S2.

We consider that the manuscript is now suitable for publication in PeerJ.

Dr. Omar Flores

Postdoctoral researcher

University of Antwerp

On behalf of all authors.

## Reviewer 2

Dear Stefan,

Thank you for your time reviewing our manuscript again.

*Overall, the authors updates really improved the manuscript.*
*There is only one thing, which I missed in my previous assessment.*
*"... soil structure as ecosystem engineers are predated...", this is the first mentioning of the term ecosystem engineer.*
*From ecology, ecosystem engineers are organisms that creates, significantly modifies or maintains an ecosystem.*
*Sure earthworms are doing that, but so are beavers, sphagnum mosses, kelp and some bacteria.*
*It would be much better to replace ecosystem engineer with earth worms throughout the document, since that is what you are referring to. But of course acknowledging that earth worms can be considered to be ecosystem engineers.*
*One reason I am assuming that is because you are allowing pH to directly affect the group, not sure that ants would be as affected by pH as earth worms are.*
*If not, then it has to be very clear that you in this study how you are defining ecosystem engineers.*

Regarding the concept of ecosystems engineers, it was referenced in the second paragraph of the introduction: "ecosystems engineers, sensu Jones et al. (1994)", but in order to make it clearer, we have introduced explicitly a short definition before the reference. We refer to (potentially) all ecosystem engineers within soil fauna, defined by average values between different species. And later, in the particular case of the example we provided with data from Brasschaat, we used earthworms as an example of ecosystem engineers. But we cannot refer only to earthworms in the description of the model because it is not intended to simulate only earthworms.

Moreover, in the methodology, section "Input parameters of species", lines 554-555 of the previous submitted version, we stated that "The soil fauna groups used consist of a wide range of species, for which average data are used."

We have added also another more explicit clarification in section "Model parameterization", stating that earthworm biomass is used in this parameterization as an example of ecosystem engineers. We used it because that was the available data we had, although we acknowledged in the manuscript that this was not an optimal ecosystem for earthworms.

The function introduced to simulate the pH effect on engineers (in this case on earthworms) could be easily modified changing the parameters if the model is used to simulate other engineer species that are not so sensitive to pH.

## Reviewer 3

Dear reviewer,

Thank you for your time reviewing our manuscript again and for your suggestions.

*I have mainly suggestions to improve their discussion a bit. For instance, they have three citations in total in their entire discussion. Authors should discuss their results with a little more context before the manuscript is accepted. For example, there were two recent review papers out highlighting the importance of soil predators in driving soil microbiome (Thakur and Geisen 2019, Trends in Microbiology and Erktan et al. 2020, Soil Biology & Biochemistry), which could easily be integrated in their discussion (lines 668-682). Another minor issue is the way authors claim KELYLINK as "one of the first and most ambitious attempts ....". I will be careful and tone it down- particularly "the most ambitious attempts". The authors should leave that to the readers to judge!*

We totally agree. The suggested papers are really interesting and they have been included in the discussion. We also improved the discussion section with further discussions and several additional references. And the abstract was also edited for the new version of the manuscript.

## Reviewer 4

Dear reviewer,

Thank you for your time reviewing our manuscript, and for all your detailed comments that were really helpful to improve our work. Please find our answers below:

*Language use throughout the article could use improvement. There are numerous grammatical issues throughout the text and the writing style at some points is less formal than typical for academic articles. I would recommend thorough proofreading of the text for grammar and style. In addition, at some points the text reads less as a scientific description and evaluation of the model and more as an advertisement for others to use the model.*

We have edited our writing, checking text for grammar issues and improving the style. We hope it looks better now.

*The application of the model and calibration of parameters, however, have a lot of problems and I think the model simulations overall fail to reach accepted technical standards. The model has a large number of pools (13) and a lot of parameters that are poorly constrained by site measurements. Parameter estimation was conducted using a Bayesian Markov Change Monte Carlo approach using only 9 data points, of which several were literature-based estimates rather than site estimates. The study treats these 9 data points (representing a biomass number for each of 9 pools) as 99 points by assuming steady state across years, which I think is unjustifiable pseudo-replication. Overall, I think a Bayesian calibration of a model with 13 C pools using only a total of 9 data points cannot be justified. This is borne out in the results, such as Tables 5 and 6 which show huge ranges in parameter value distributions. In addition, mean pool values (Table 5) are not at all consistent with the data values (Table 2), suggesting that the calibration was unsuccessful. For example, fungal biomass was calibrated to a value of 15 g C m-3 (Table 2) but in model simulations has a mean value of 200 (Table 5).*

We want to clarify that this is not a model-application study but rather the description and representation of a new-model concept. Obviously, for a full model application we would require more data and we would investigate the ecosystem and the responses in more detail and improve on the calibration. The dilemma with new model publications is that the manuscripts inevitably become too long, and the focus tends to deviate from the model description to the application. For future users of the model, the model itself is important, not the details of the Brasschaat forest. We deliberately chose not to couple the soil model to a vegetation model to emphasize the patterns observed when playing with the different critical factors of the soil model (e.g. clay content, predators removal, etc.), but obviously the ultimate goal is to couple KEYLINK to ecosystem models.

About the calibration, we used 11 data points from 11 C pools (not 9) as reference data, in order to calibrate 9 parameters (the 9 gmax). Maybe the misunderstanding with the 9 comes because those 11 pools were mentioned as "litter, SOM, and the 9 functional groups in the food web" (lines 534-535 in previous version), or because those 11 points were replicated 9 times assuming steady-state over 9 years. We didn't use two of the C pools, the roots and the $CO_2$. The reason for this was, in the case of roots, to simplify the vegetation belowground allocation to roots, which was treated as a constant C input, so it makes no sense to calibrate towards that for the moment; on the other hand, $CO_2$ is another pool that serves only to collect all the C outputs from the soil, so it's just the C released to the atmosphere and not a C pool that remains and interacts with the system. For those reasons, we only used the other 11 C pools for the calibration. We consider this decision is amply justified.

We agree on the poor constrain of parameters, which is due to the scarce data available from a single site and for all the parameters needed, as we discussed in the manuscript. That is also why we needed to base some parameters on literature.

Regarding the pseudo-replication, we agree with the referee, but we want to highlight again that our goal here was not to obtain a realistic simulation of the forest of Brasschaat, rather than to use some of the data from Brasschaat to create a hypothetical ecosystem example that illustrates the model outputs. That is why we just assumed a steady-state and calibrated the model towards it, which is also a common procedure for model calibrations where the model is calibrated towards the initial state. This is not a paper about Brasschaat, but about the model, so taking into account the scarce data available, we think it is reasonable to hold some assumptions, until more data allows to conduct a more realistic calibration for any particular ecosystem. We explained it in the first paragraph of the section "Calibration for Brasschaat pine forest", but we have added there a more explicit explanation that the calibration was conducted towards a partially assumed data that does not fully fit with reality in Brasschaat forest.

*It does not seem that there is enough data available for the study's field site to actually constrain the parameters of the model, so it's hard for me to say what the best path forward might be. I think that for this study to be successful the calibration procedure needs to be completely re-conceived and rewritten in a way that is consistent with the availability of data to constrain the model's parameters. It is extremely important to evaluate the calibration procedure itself, which clearly failed in the results presented. Careful examination of posterior parameter distributions and the pattern of the MCMC results is important for establishing whether the parameterization procedure actually converged in a useful way.*

As explained above, the aim of this study was not to simulate C budgets of Brasschaat. The aim of this study, as it is well explained throughout the whole manuscript, was to present to the scientific community a first preliminary version of a new concept model that hopefully will serve to challenge current state-of-the-art soil modelling. But we are aware that to do that we will need to improve the calibration of the model in the future, using more complete databases that take into account all the elements needed to calibrate KEYLINK, which, on the other hand, are currently extremely scarce. We, therefore, hope that by presenting this concept model that challenges the current way of simulating soil biochemical cycling, we will stimulate that future studies will also be designed to take into account the pools and functional groups needed to calibrate KEYLINK.

We cannot calibrate the KEYLINK model for Brasschaat without including and calibrating an above-ground growth model for the vegetation which is far beyond the scope of this manuscript and would not increase the understanding of the model functioning because the models would interact.

*Perhaps a more theoretical analysis of the model behavior, using reasonable parameter values and focusing on understanding interactions among different pools in more detail,*

*would be one path forward. Overall I think this model is a good example both of the process insights that can be gained by increasing model complexity and an example of the difficulty of getting meaningful results from an increasingly complex model with parameters that are difficult to constrain.*

Indeed that was our goal from the start, to focus on the model behavior and not on the specific site of Brasschaat. We have tried to make this clearer in the manuscript throughout. We agree that the complexity of the model makes it hard to obtain a very successful calibration, as our results showed.

*One of the most promising aspects of the model is the connection between faunal engineers and soil structural changes. It seems like a missed opportunity to run the model only at a site with very low faunal engineer biomass, where this effect cannot be investigated fully.*

About the missed opportunity, we expect to run the model for more sites in the future, but that will require very detailed data of those sites and for this version we used the site from which we have more data availability. We are currently working on improvements for the simulation on engineers effect on soil structure, so we will address that opportunity better in next versions of the model.

Our goal is to have the equations published with enough application to understand the goal and the limitations, so in future model runs we can refer to this publication.

*The results section of the manuscript is very problematic. The poor design of the model calibration and the complexity of the model itself produced results that are difficult to interpret and inconsistent with reality. The interpretations of the results in the manuscript take an optimistic view that is not consistent with the actual results. Model simulations produced a very wide range of outcomes as evident in Table 5. Biomass time series (Figure 3) look very unstable and are characterized by short-term spikes in biomass of some pools, including huge variability in bacterial biomass, that is not realistic for an ecosystem-scale soil model. Because only the mean of multiple simulations is shown, it is difficult to tell how these time series varied among simulations (within the same scenario) but I expect the variability is very high, calling into question many of the interpretations of the results. It would help to see the variability in simulations behind Fig 3. I would not be surprised if individual simulations within each scenario were extremely variable due to parameter uncertainty (consistent with Table 5 and 6).*

We partially disagree with the referee. Although some results don't seem very realistic, microbial biomass is a pool very unstable in real conditions that depends very much on water availability (Blackwell *et al*., 2010; Zhao *et al*., 2010). If models predict a change

in water availability to the system (in this case due to an increase in engineer presence and hence a peak in water infiltration to soils) we do not see why microbial biomass cannot show non-linear increases, as it has been amply observed in literature. Of course, there is not available data to test how realistic these modeled fluctuations in functional groups such as engineers are, but this is just a very theoretical example, that will be hopefully tested in the near future. We have added this to the discussion.

*The analysis of the results is very short and quite shallow and does not address the key aspects of the different scenarios that were simulated. There is not really a meaningful analysis of the model simulations beyond a cursory, qualitative description. The results and discussion seem to start from the assumption that the model will be useful because of the processes it includes, and do not reflect an actual objective analysis of the simulation results. There is a lot of complexity in the model results which is mostly overlooked. I think to meet scientific standards the results and discussion would need to be completely rewritten to reflect an objective assessment of the model results.*

*Overall, the results and discussion are more an argument trying to justify the value of the model's structure than a real analysis of the model outcomes. The analysis of the calibration procedure does not take a serious critical view of the possibility that the Bayesian approach might not give strong constraints on the model parameters. Part of the value of a Bayesian analysis is that it provides estimates of parameter uncertainties, and insights about whether parameters could be well constrained by available data, and it seems that the analysis did not take advantage of this but rather started with the assumption that the model could be well constrained despite the limited measurements that were used.*

Again, we agree that the purpose of this manuscript is to describe the model and not to model the forest site of Brasschaat. We hope that by making this more clear from the start our readers will no longer 'expect' a detailed analyses of the Brasschaat forest soil functioning. We did add a more critical discussion on the calibration.

*Line 31: I suggest starting the abstract with a sentence about the scientific context or knowledge gap that motivate the model before jumping into the model description*
Done.

*Lines 34-37: This reads as more of an advertisement for the model than a statement about the science. In this study, KEYLINK was not coupled to another ecosystem model so it does not seem relevant to the study to say how it could be coupled.*
Those sentences have been rewritten in the revised version of the manuscript.

*Line 50: The model was not actually compared with a first-order model, so there isn't*

*evidence that it was actually a more successful approach.*
The mention to first-order model has been deleted.

*Line 59: Century and RothC are not the oldest soil models that exist. It would be more accurate to just say that they are widely used*
Done.

*Line 102: I would start the methodology section with an overview of the whole model and how the pieces relate to each other as shown in Fig 1.*
We have added a few lines introducing the methodology, but without explaining the relationships among pieces, because that would be to repeat what is already in the introduction section.

*Line 114: If earthworms eat all soil, do they have access to all pore sizes, or only the bacterial pores? It is not clear from the description*
Earthworms feed on SOM from all pore sizes except inaccessible pores, because SOM is physically protected among soil particles in those pores. This has been included more explicitly in the manuscript.

*Line 127-129: The statement about macroporosity should have a citation to literature supporting it. And this statement does not seem to fit with this section since none of the other pore size classes are described in terms of laboratory measurements.*
That statement has been removed following the reviewer's advice that it didn't fit in that section.

*Line 139: Equation 1 should have a + between the terms in the denominator, not a -*
The referee is right. It has been corrected.

*Line 153-155: Is evaporation assumed to be equal to potential evapotranspiration? This does not seem realistic since it ignores source limitation of evaporation as well as boundary layer and conductance effects that limit actual evapotranspiration*
We did not describe this well. We did use potential evaporation in our run, but it is not part of the model. We have moved this to line 545 where we describe the test run we did. In the model, in contrast, evapotranspiration is an input which should come from data or from a vegetation model (as is now explained in the text) and which is influenced by multiple factors as stated by the reviewer.

*Line 199: Equation 9 should include terms for modifiers to gmax (temperature, pH, etc). The sum notation should indicate the index that is being summed over. The role of*

*excreted faeces also needs to be in this equation, otherwise it states that all of the substrate is being converted to biomass growth which is untrue.*

The sum notation has been corrected, adding n as a subscript representing each C source, and N as the total number of fluxes (one per source).

Regarding the modifiers to gmax, they are explained in the sections below, so it would be confusing to introduce them already there. gmax is explained as a rate at this step, and we think that the explanation of how it is calculated makes more sense after the explanation of all the modifiers instead of before them. On the other hand, about faeces, that was already explained in lines 204-205; G is not the consumed biomass, but the assimilated biomass from the consumed source. Faeces are already included in equation 13, so the inclusion of them here would make to count them twice in equation 10.

*Line 205-207: An equation should be shown or referenced for fa*

We added a reference showing that fa was explained with equations 14-16.

*Line 208: Physical recalcitrance is far from a novel concept, and has been included in conceptual and numerical models prior to KEYLINK (e.g., MIMICS, and the passive pool in CENTURY)*

We deleted that adjective.

*Line 218: The model simulates increases in biomass of pools, not in population number*

The term number has been replaced by biomass.

*Line 225: It's confusing to refer to temperature sensitivity in growth when gmax as presented so far does not include temperature dependence (Eq. 9). Adding these factors to Eq. 9 would make this clearer.*

Temperature modifier ($m_T$) has been added to equation 25, in which all growth modifiers are applied.

*Line 235: This equation states that predation rate depends only on the predator's total growth rate from all substrates, and not on the biomass of the prey. This does not make sense. If a predator feeds on multiple types of prey, then this suggests that all are predated at the same rate regardless of their different biomass amounts. This equation would make sense if it refers to the fraction of a predator's growth from a single prey type, but that is not what the equation actually shows.*

We disagree with referee's comment, in that equation $G_{pred}$ is the growth of the predator, being G calculated with equation 9 (we added a note clarifying this in the new version), and in that equation the biomass of the prey (source) is included.

*Line 248: availability was f_a in Eq. 9, but is noted as (a) here*

Corrected as f_a.

*Line 254-312: This description would be easier to follow if it was moved to be right after Eq. 9 (which describes gmax), or if there were a statement after Eq. 9 stating that modifiers to gmax were described below. Either way, Eq. 9 should show that gmax is modified by additional factors.*

It is complicated to organize those sections because in any case there would be parts mentioned but not fully explained after other parts also mentioned above. So we added a statement that several modifiers are applied to gmax and they are described below. We thank the referee for the advice.

*Line 259: Density-dependent microbial turnover should be a modifier on microbial death rate, not on growth rate*

Yes, of course. But we just explained that some potential add-ons, as for example density-dependent microbial turnover, are not included yet. It is not used for growth rate. Maybe it is confusing that we mentioned it in that section, so in order to clarify it, we have modified that sentence to show that density-dependent microbial turnover could be added as a modifier to death rate.

*Line 271: This equation has a discontinuity in it at T=Tmax, where mT goes directly from 1 to 0. This doesn't make much biological sense*

It is true that a more realistic function would decrease gradually, which could be done with a few more parameters. We chose to simplify that since a proper function would need more data, and that could be added to the model replacing this function with any other if enough data is available for it.

*Line 280-281: The text should reference literature supporting the pH effects on bacteria and fungi*

We have added in the new version of the manuscript two references supporting those effects.

*Line 283-284: These equations do not seem correct. gmax would approach infinity as pH approaches the threshold in each case. For example, at pH=8.01, mpH for fungi would be 10. An exponential function would work better. Also, Eq. 18 defines a declining fungal growth at high pH and constant fungal growth at lower pH (< 8), while the text above refers to an increasing fungal growth at low pH.*

We have added a sentence clarifying that if mpH goes above 1, it is replaced by 1, which is the maximum. Regarding the equations, it is true that they were not correctly explained; we have indicated that there is a precision of one decimal, so >8 stands for

≥8.1, but we changed it to make it clearer. About the other issue, although we discussed that fungi growth increases at low pH, in the equations we simplified it by making it constant at its maximum below its threshold. On this there are also issues on data availability and overparameterization, so we decided to apply that simplification. Of course the function could be easily replaced if enough data is available at a site to properly parameterize all the responses of microbial growth to different pH values.

*Line 287: It is not clear why there is a linear response of engineer saprotrophs to pH but a 1/pH response for bacteria and fungi. Is there some literature support for this choice?*
We have added several references from literature supporting a linear response to pH of engineer saprotrophs within a range of values, contrasting with different responses for bacteria and fungi.

*Line 291-293: This sentence does not make grammatical sense and needs to be rewritten.*
The sentence has been rewritten to make it clearer.

*Line 294: mrec should be defined when it is introduced*
mrec was introduced in the previous sentence as a modifier depending on recalcitrance.

*Line 303: Why was a linear equation chosen here and a power law above?*
We consider those two functions to be different, since constrain by C:N ratio of decomposition on labile litter should not completely stop decomposition but adjust the decomposition rate, while recalcitrant fraction of litter could remain almost constant for long periods (as observed experimentally). The linear equation for recalcitrance modifies the fraction of litter affected by decomposition, which could becomes (1-Rec_lit) in case pmrec=1, i.e. the complementary of Rec_lit, which is the labile fraction of litter. That second equation determines if the recalcitrant fraction of litter remains stable or if it is affected by decomposers partially or even totally. This explanation has been added to the manuscript.

*Line 314: This section is not really about closing the C budget, it is about the fraction of decomposed material that is converted to faeces (which also needs to be included in Eq. 9)*
The title of that section has been changed in the new version of the manuscript.

About faeces, we have already explained (in our answer to comment on line 199) why faeces are not included in equation 9. Faeces are included in equation 13 for predation.

*Line 325 and 328: Recalcitrance is not a conserved mass quantity and does not have a budget*
*Line 335: Based on the text, N limitation should be modifying r rather than gmax*
*Line 336: This equation states that growth rate is positively correlated with C:N of SOM, the opposite of the statement here that growth slows with lower N*
*Line 348-349: What specific parameter is "twice as recalcitrant" referring to?*

We have removed that section from the manuscript, because in the presented example the simulations were not using that. Therefore, we will present those equations on a future version.

*Line 360-361: These units do not make sense.*
l m$^{-3}$ stands for liter per cubic meter of soil, and l g C eng$^{-1}$ (or l / g C eng) are the liters per gram of C in engineer pool. The units of engineer biomass have been added to make it clearer.

*Line 363-364: Eq. 31 burrow volume as being directly proportional to engineer biomass, in an instantaneous way. It does not make sense to pair this with a turnover rate. If rates are defined, then burrow volume needs to have both a formation and a turnover rate.*
There was another part using parameters of the following section, but the equation 31 has been amended including also the turnover rate, with a reference indicating that some of the parameters are explained in the following section.

*Line 369: Should these units be m3/m3?*
It could be, but l m$^{-3}$ is also correct, and we keep the same volume units to make it easier to link everything in the same units, including the hydrology.

*Line 395: These units are also incorrect*
We thank the referee for noticing that a "m$^{-3}$" was missing. We have added it.

*Line 398: Can layer thickness change in the model? Based on other equations, it does not seem like it, so this statement is confusing*
That refers to a change in density. It has been changed to make it less confusing.

*Line 423-446: This section does not actually specify how much DOM is leached*
We have revised and edited that section.

*Line 472-474: Since KEYLINK has not actually been modified or calibrated for*

*different ecosystems or coupled to any other ecosystem models, this statement is unsupported. I think this whole paragraph (except for the Github link) could be removed since it is mostly an advertisement of the model and not a scientific statement.*

That statement has been erased. However, the paragraph contains practical details about the options that can be adapted to use the model in different ways. As we explained before, our goal here is not to present a modelling for a specific ecosystem or use, but the development and usefulness of the model itself (despite there is still room for many upgrades), so we consider interesting to state clear enough that some of the parameters and options as they are used in this case are just an example but can be adapted for many other purposes, because they have been designed that way.

*Line 480-481: This text should specify why multiple runs are needed (to explore parameter space). generally, I think this would be specific to the uncertainty in model parameters rather than general to the model itself, so I'm not sure it makes sense as a general recommendation.*

That statement has been erased.

*Lines 506-531: I think these lines could be removed. It is not necessary to describe the basics of how Bayesian parameter estimation works. Previous literature could be cited instead.*

Those lines have been removed following the reviewer's advice.

*Line 524-525: This is the only mention of a drylands version of the model, and seems irrelevant to the rest of the manuscript.*

That is true, we forgot to delete that mention from another version. It has been removed (together with those lines). Thanks for noticing.

*Line 536: Assuming that 9 data points are equivalent to 99 data points by assuming pool values are constant is unjustifiable. I think this is a fatal flaw of the parameterization approach.*

We didn't assume pools to be constant, but to be stable, fluctuating seasonally but with similar interannual values, so we calibrated it towards a steady-state. We have deeply discussed this in our second answer to your comments, and we hope it is clearer now.

*Line 550-552: This is not specific to this study and could be removed.*

We disagree; as previously, this is not "a study" of a particular case, but a presentation of the model and how it can be used for different applications. We think that a lack of explanations about how other people can apply the model for their own purposes would weaken the perception of the model as a tool.

*Line 575: R is respiration rate predicted by the model. It is not a parameter. Should this be r?*

Indeed, R in that paragraph should be r. It has been corrected.

*Line 578: This calibration procedure used 9 data points to constrain 9 model parameters. There is much too little data for this to be a workable approach.*

In that line we only mention that we calibrated 9 parameters. The data points are discussed in the previous section, and we have already discussed here why we made that assumption.

*Line 582: There are not 81 data points. There are 9 data points.*

We were just expressing in that line that for 9 parameters we need nine squared (i.e. 81) data points.

*Line 584: "no measurements were available" — measurements of what exaclty?*

Measurements of growth rates. We agree it was not clear enough, so we have clarified that.

*Lines 614-617: Clay, litter recalcitrance, and litter C:N are not included in Table 1*

All the parameters that were changed for the alternative scenarios are explained in Supplemental File S2, as it was indicated in the manuscript.

*Line 621: What is LHS?*

We forgot to remove that mention. In the first version of the manuscript we used a Latin Hypercube Sample (LHS) from the posterior distribution of the calibration, but for the second version we changed that. It has been removed. Thanks again for noticing those errors.

*Line 606-624: Was the model calibrated to steady state for each scenario? Or did it use the baseline parameters for all scenarios? If the second is true, then the model was likely out of steady state for other scenarios, making the results unreliable*

We used only one calibration of the model for the basal scenario. Although realistic scenarios would require specific calibrations, different calibrations would also produce more differences attributable to changes in the other parameters than to the own effect of the studied parameter in each case. For the same reason we simplified the vegetation instead of including a proper simulation of the entire ecosystem, as we have explained: we are not showing how reliable scenarios could happen in Brasschaat, but how the model reacts to some changes, using some data from Brasschaat as well as some

simplified assumptions. All changes in parameters for the different scenarios have been detailed in Supplemental File S2.

*Line 630-634: Based on Table 5, it looks like the model parameters were not well constrained at all. The +/- values don't make much sense as they include negative biomass for most pools. I imagine the distributions are skewed, so it would make more sense to show a figure with actual distributions rather than report a standard deviation that is not a good measure of actual variance. Showing posterior parameter distributions would provide better information about whether the MCMC approach was actually effective at constraining parameters.*

We have edited Table 5, which now shows only the average values, and we added a new Table 6 including maximum and minimum values of parameter distributions.

*Line 635: What is meant by "relatively uncoupled?" They were poorly correlated with each other?*

It can be seen that their fluctuations do not follow the same patterns, despite they are influenced by their predators and all functional groups are indirectly linked through the food web, but still those three are an example of uncoupled patterns. It could be said also that they were poorly correlated with each other.

*Line 637-638: The behavior of bacteria in the model is clearly unrealistic, with a huge biomass spike at the beginning of the simulation followed by death of basically the entire bacterial community.*

Yes, and we already discussed that in lines 638-644.

*Line 642: "bacteria would profit most from a rapidly changing environment" This doesn't make much sense either in the model or in real life. Typical soils have large bacterial populations whether they are rapidly changing or not*

Bacterial populations are large in soils, and they have great biodiversity, which together with their faster growth rates should allow bacteria (either some species or others among all their diversity) to adapt faster and benefit from new environmental conditions. We have added also that under some unrealistic simulated conditions fungi could be displacing bacteria by competitive exclusion, to make that explanation clearer.

*Line 645: Looking at Figure 3, the opposite of this statement is true. C pools seem very unstable and are characterized by spikes and large fluctuations which suggests to me that the model is poorly balanced. One scenario lost 75% of SOM in the first year!*

We stated that stability is reached after the first years (implying that first years are quite unstable). After ca. 1000 days of simulation, C pools tend to fluctuate around the same average values for the rest of the simulation, which is some stability. We have added a nuance to emphasize that stability is partial.

Of course peaks are not very realistic, but as we have discussed previously, it cannot be expected to find very realistic fluctuations without a proper simulation of the entire ecosystem including the vegetation, which was not included in those simulations. For that reason we also insisted several times through the paper in the capability of the model to be coupled to other models, not as a simple advertisement, but to make it clear that KEYLINK is a puzzle piece to build a more complex integral representation of the whole ecosystem.

*Line 651: Different soil layers should have the same long-term average temperature, although shallower soil layers would be expected to have wider fluctuations*
We have edited that statement.

*Line 658-659: The study included six different model scenarios, but only one is discussed and only in one sentence here.*
We have added further discussions of the scenarios.

*Figure 2: The diagram should show which pools are external to the model (tree shoots, litter, CO2) and which are actually model pools. From the text, it seems like the SOM pools shown (different POM sizes, DOC) are not actually in the model, so it is misleading to show these as model components in the diagram. The diagram should be consistent with the model that was actually used in the study.*
The Figure 2 has been replaced by an improved version.

*The colors of the lines in Figures 3 and 4 are very difficult to tell apart.*

We agree. We changed them from a first version with clearer colours following the Journal's advice, but we have improved the legend again. We hope now it is easier to see the different lines both by color or by line shape (for colour-blind people).